# Tikhonov Regularization is Optimal Transport Robust under Martingale Constraints

**Jiajin Li**    **Sirui Lin**    **José Blanchet**
Stanford University
{jiajinli, siruilin, jose.blanchet}@stanford.edu

**Viet Anh Nguyen**
Chinese University of Hong Kong
nguyen@se.cuhk.edu.hk

## Abstract

Distributionally robust optimization has been shown to offer a principled way to regularize learning models. In this paper, we find that Tikhonov regularization is distributionally robust in an optimal transport sense (i.e., if an adversary chooses distributions in a suitable optimal transport neighborhood of the empirical measure), provided that suitable martingale constraints are also imposed. Further, we introduce a relaxation of the martingale constraints which not only provides a unified viewpoint to a class of existing robust methods but also leads to new regularization tools. To realize these novel tools, tractable computational algorithms are proposed. As a byproduct, the strong duality theorem proved in this paper can be potentially applied to other problems of independent interest.

## 1  Introduction

Regularization is an important tool in machine learning which is used in, for instance, reducing overfitting [23]. Recently, ideas from distributionally robust optimization (DRO) have led to a fresh viewpoint on regularization precisely in connection to overfitting; see, e.g., [7, 2, 20, 10, 29, 24, 25, 5] and the references therein.

In these references it is shown that many standard regularization-based estimators arise as the solution of a min-max game in which one wishes to minimize a loss over a class of parameters against an adversary that maximizes the out-of-distribution impact of any given parameter choice, that is, the adversary perturbs the empirical distribution in a certain way. The choice of adversarial distributions or perturbations in DRO is often non-parametric thus providing reassurance that the decision is reasonably robust to a wide range of out-of-distribution perturbations. For example, one such non-parametric choice is given by employing optimal transport costs [31] to construct a so-called distributional uncertainty set (e.g. a Wasserstein ball around the empirical distribution) for the adversary to choose. [27] shows that optimal transport-based DRO (OT-DRO) is closely related to adversarial robustness in the sense of steepest gradient loss contamination. This can be further explained by OT-DRO's hidden connection with generalized Lipschitz regularization [7]. Thus, understanding if a well-known regularization technique is actually distributionally robust and in what sense, allows us to understand its out-of-distribution benefits and potentially introduce improvements.

In this paper, we introduce a novel set of regularization techniques which incorporate martingale constraints into the OT-DRO framework. Our starting point is the conventional OT-DRO formulation. The conventional OT-DRO formulation can generally be interpreted as perturbing each data point in such a way that the average size perturbation is less than a given budget. In addition to this

conventional formulation, we will impose a *martingale constraint* in the joint distribution of the empirical data and the resulting adversarially perturbed data.

*Why do we believe that the martingale constraint makes sense as a regularization technique?* It turns out that two random variables $X$ and $\bar{X}$ form a martingale in the sense that $\mathbb{E}[\bar{X}|X] = X$ if and only if the distribution of $\bar{X}$ dominates $X$ in convex order [30]. In this sense, the adversary $\bar{X}$ will have higher dispersion in non-parametric sense than the observed data $X$ but in a suitably constrained way so that the average locations are preserved. This novel OT-DRO constrained regularization, we believe, is helpful to potentially combat conservative solutions, see [16]. Moreover, by allowing a small amount of violation in the martingale property, we can control the regularization properties of this constraint, thus obtaining a natural interpolation towards the conventional OT-DRO formulation and potentially improved regularization performance. We point out that related optimal transport problems with martingale constraints have been studied in robust mathematical finance [1, 8].

Consider, for example, the linear regression setting with the exact martingale constraints, which means that for any given observed data point, the conditional expectation of the additive perturbation under the worst-case joint distribution equals zero. Surprisingly, we show that the resulting martingale DRO model is *exactly* equivalent to the ridge regression [18] with the Tikhonov regularization. To the best of our knowledge, this paper is the first work to interpret the Tikhonov regularization from a DRO perspective showing that it is distributionally robust in a precise non-parametric sense. In stark contrast, it is well-known that the conventional OT-based DRO model (*without* the martingale constraint) is identical to the regularized square-root regression problem [2]. Therefore, introducing an additional power in norm regularization (i.e., converting square-root regression to Tikhonov regularization) can be translated into adding martingale constraints in the adversarial perturbations thus reducing the adversarial power. A natural question that arises here is whether we can interpolate between the conventional DRO model and the Tikhonov regularization, and further improve them.

We will provide a comprehensive and positive answer to the above question in this paper. The key idea here is to relax the equality constraint on the conditional expectation of the adversarial violation and thus allow a small perturbation of the martingale property to gain more flexibility of the uncertainty set. This idea leads to another novel model, termed the perturbed martingale DRO in the sequel. Intuitively, if the relaxation is sufficiently loose, the perturbed martingale DRO model will reduce to the conventional DRO, which is formally equivalent to setting an infinite amount of possible violations for the martingale constraint. By contrast, if no violation is allowed, the perturbed martingale DRO will automatically reduce to the exact counterpart — Tikhonov regularization. As a result, we are able to introduce a new class of regularizers via the interpolation between the conventional DRO model and the Tikhonov regularization.

Furthermore, such insightful interpolation also works for a broad class of nonlinear learning models. Inspired by our extensive exploration of linear regression, the developed martingale DRO model can also provide a new principled adversarial training procedure for deep neural networks. Extensive experiments are conducted to demonstrate the effectiveness of the proposed perturbed martingale DRO model for both linear regression and deep neural network training under the adversarial setting.

We summarize our main contributions as below:

- We reveal a new hidden connection in this paper, that is, Tikhonov regularization is optimal transport robust when exact martingale constraints (i.e., convex order between the adversary and empirical data) are imposed.

- Upon this finding, we develop a new *perturbed* martingale DRO model, which not only provides a unified viewpoint of existing regularization techniques, but also leads to a new class of robust regularizers.

- We introduce an easy-to-implement computational approach to capitalize the theoretical benefits in practice, in both linear regression and neural network training under the adversarial setting.

- As a byproduct, the strong duality theorem, which is proved in this paper and is used as the main technical tool, can be applied to a wider spectrum of problems of independent interest.

## 2   Preliminaries

Let us introduce some basic definitions and concepts preparing for the subsequent analysis.

**Definition 2.1** (Optimal transport costs and the Wasserstein distance [21, 31])**.** *Suppose that $c(\cdot, \cdot) :$ $\mathbb{R}^d \times \mathbb{R}^d \to [0, \infty]$ is a lower semi-continuous cost function such that $c(X, X) = 0$ for every $X \in \mathbb{R}^d$. The optimal transport cost between two distributions $\mathbb{Q}$ and $\mathbb{P}$ supported on $\mathbb{R}^d$ is defined as*

$$D(\mathbb{Q}, \mathbb{P}) \triangleq \min_{\pi \in \mathcal{P}(\mathcal{X} \times \mathcal{X})} \left\{ \mathbb{E}_\pi \left[ c(\bar{X}, X) \right] : P_1 \pi = \mathbb{Q}, \ P_2 \pi = \mathbb{P} \right\}.$$

*Here, $\mathcal{P}(\mathcal{X} \times \mathcal{X})$ is the set of joint probability distribution $\pi$ of $(\bar{X}, X)$ supported on $\mathcal{X} \times \mathcal{X}$ while $P_1 \pi$ and $P_2 \pi$ respectively refer to the marginals of $\bar{X}$ and $X$ under the joint distribution $\pi$.*

If $c(X, \bar{X}) = \|\bar{X} - X\|$ is any given norm on $\mathbb{R}^d$, then $D$ recovers the Wasserstein distance [31]. In this paper, we are interested in a flexible family of functions for the computational tractability, so called the Mahalanobis cost functions in the form of $c(\bar{X}, X) = (X - \bar{X})^\top M (X - \bar{X})$, where $M$ is a $d$-by-$d$ positive definite matrix.

Next, we consider the conventional OT-DRO problem:

$$\min_\beta \mathbb{L}_\beta(\widehat{\mathbb{P}}, \rho), \quad \text{where} \quad \mathbb{L}_\beta(\widehat{\mathbb{P}}, \rho) \triangleq \begin{cases} \sup_\pi & \mathbb{E}_\pi [\ell(f_\beta(\bar{X}))] \\ \text{s.t.} & \pi \in \mathcal{P}(\mathcal{X} \times \mathcal{X}) \\ & \mathbb{E}_\pi \left[ c(\bar{X}, X) \right] \leq \rho, \ P_2 \pi = \widehat{\mathbb{P}}, \end{cases} \tag{2.1}$$

where $\widehat{\mathbb{P}} \triangleq \frac{1}{N} \sum_{i=1}^N \delta_{X_i}$ is the empirical distribution. Using Definition 2.1, we have $\mathbb{L}_\beta(\widehat{\mathbb{P}}, \rho) = \max_{\mathbb{Q}:D(\mathbb{Q},\widehat{\mathbb{P}})\leq\rho} \mathbb{E}_\mathbb{Q}[\ell(f_\beta(\bar{X}))]$, which is the worst-case expected loss under all possible distributions around the empirical measure $\widehat{\mathbb{P}}$ at most $\rho$ with respect to the OT distance. It is well-known that under appropriate assumptions, the DRO problem (2.1) is equivalent to the regularized square-root regression problem.

**Proposition 2.2** ([2, Proposition 2.])**.** *Suppose that (i) the loss function $\ell(\cdot)$ is a convex quadratic function, i.e., $\nabla^2 \ell(\cdot) = \gamma > 0$, where $\gamma$ is a constant, (ii) the feature mapping $f_\beta(\bar{X}) = \beta^\top \bar{X}$ is linear, and (iii) the ground cost $c$ is the squared Euclidean norm on $\mathcal{X} = \mathbb{R}^d$. Then*

$$\mathbb{L}_\beta(\widehat{\mathbb{P}}, \rho) = \left( \sqrt{\mathbb{E}_{\widehat{\mathbb{P}}}[\ell(f_\beta(X))]} + \sqrt{\rho}\|\beta\|_2 \right)^2.$$

We also present the strong duality result for a general class of optimal transport based DRO models with martingale constraints. This result serves as our main technical tool for reformulating the DRO models, and it can also be applied to other semi-infinity structured DRO models, which could be of independent interests. We consider the primal problem

$$\begin{array}{ll} \sup_\pi & \int_{\mathcal{X} \times \mathcal{X}} f(\bar{X}) \, \mathrm{d}\pi \\ \text{s.t.} & \pi \in \mathcal{P}(\mathcal{X} \times \mathcal{X}) \\ & \int_{\mathcal{X} \times \mathcal{X}} c(\bar{X}, X) \, \mathrm{d}\pi \leq \rho, \ \ P_2 \pi = \widehat{\mathbb{P}} \\ & \mathbb{E}_\pi[\bar{X}|X] = X \ \ \widehat{\mathbb{P}}\text{-a.s.} \end{array} \tag{Primal}$$

and its associated dual form

$$\inf_{\substack{\lambda \in \mathbb{R}_+ \\ \alpha_i \in \mathbb{R}^d \ \forall i}} \lambda\rho + \sum_{i=1}^N \alpha_i^\top X_i + \frac{1}{N} \sum_{i=1}^N \sup_{\bar{X}} \left[ f(\bar{X}) - \alpha_i^\top \bar{X} - \lambda c(\bar{X}, X_i) \right]. \tag{Dual}$$

Here, $f : \mathcal{X} \to \mathbb{R}$ is upper semi-continuous and $L^1$-integrable. The next theorem states the strong duality result linking these two problems.

**Theorem 2.3** (Strong duality)**.** *Let $\widehat{\mathbb{P}} \triangleq \frac{1}{N} \sum_{i \in [N]} \delta_{X_i}$ be the reference measure. Suppose that (i) every sample point is in the interior of the cone generated by $\mathcal{X}$, i.e., $X_i \in \text{int}(\text{cone}(\mathcal{X})) \ \forall i \in [N]$, and (ii) the ambiguity radius $\rho > 0$. Then the strong duality holds, i.e., $\text{Val}(\text{Primal}) = \text{Val}(\text{Dual})$.*

**Remark 2.4.** *At the heart of our analysis tools is the abstract semi-infinite duality theory for conic linear program [26, Proposition 3.4]. Notably, it will be tricky and subtle to reformulate our problem into the standard form and further carefully check the general Slater condition. Moreover, we indeed fill the technical gap in [19, Theorem 4.2].*

**Notation.** We use $\mathbb{S}^d_{++}$ to denote the set of $d$-by-$d$ positive definite matrices and $\|X\|_M \triangleq \sqrt{X^\top M X}$ for any $X \in \mathbb{R}^d, M \in \mathbb{S}^d_{++}$; if $M$ is the identity matrix, then we omit $M$ and write $\|X\| \triangleq \sqrt{X^\top X}$. We use $\delta_X$ to denote the Dirac measure at $X$ and let $\widehat{\mathbb{P}} \triangleq \frac{1}{N} \sum_{i=1}^N \delta_{X_i}$ be the empirical measure constructed from sample $\{X_1, \ldots, X_N\}$. We use $\mathbb{E}_P$ to denote the integration over $P$: $\mathbb{E}_P[f(X)] = \int_{\mathcal{X}} f(X)\mathrm{d}P$. Specifically, for $(\bar{X}, X)$ following joint distribution $\pi$, $\mathbb{E}_\pi[c(\bar{X}, X)] = \int_{\mathcal{X} \times \mathcal{X}} c(\bar{X}, X)\mathrm{d}\pi, \mathbb{E}_\pi[f(\bar{X})] = \int_{\mathcal{X} \times \mathcal{X}} f(\bar{X})\mathrm{d}\pi$. We use $\ell(\cdot)$ to denote the loss function applied to the parametrized feature mapping $f_\beta$. Let $\nabla\ell(\cdot), \nabla^2\ell(\cdot)$ be the first and second order derivative of $\ell(\cdot)$ respectively. Notably, we use $\mathbb{L}_\beta(\widehat{\mathbb{P}}, \rho)$ to denote the objective function of the conventional DRO model (2.1); we use $L_\beta(\widehat{\mathbb{P}}, \rho)$ to refer to the exact martingale DRO model (3.1); we use $\mathcal{L}_\beta(\widehat{\mathbb{P}}, \rho, \epsilon)$ to refer to the perturbed martingale DRO model (3.2).

## 3 Tractable Reformulations

In this section, we introduce an optimal transport-based DRO model with the exact martingale constraint at first. That is, on top of the vanilla DRO model [3], we add an additional martingale equality constraint on its coupling. It is surprisingly interesting to find out that the resulting DRO approach is equivalent to empirical risk minimization with Tikhonov regularization. Naturally, we can relax the equality constraint and thus allow a small violation of the martingale property to enrich the uncertainty set. Formally, by sending violation size to infinity in the martingale constraint, our relaxation allows to interpolate between the conventional DRO formulation (i.e. with no martingale constraints) and Tikhonov regularization (which involves exact martingale constraints). Therefore, this relaxation further leads to a new class of regularizers in a principled way, which improves upon Tikhonov regularization as we show in our experiments.

**Assumption 3.1.** *The following assumptions hold throughout.*

   *(i) The ground cost $c(\cdot, \cdot)$ is the Mahalanobis cost with the weighting matrix $M \in \mathbb{S}^d_{++}$,*

   *(ii) The domain $\mathcal{X}$ is unconstrained, i.e., $\mathcal{X} = \mathbb{R}^d$.*

### 3.1 Optimal Transport-based DRO with Martingale Constraints

To start with, we investigate the exact martingale DRO problem:

$$
\min_\beta L_\beta(\widehat{\mathbb{P}}, \rho), \qquad \text{where} \quad L_\beta(\widehat{\mathbb{P}}, \rho) \triangleq 
\begin{cases}
\sup_\pi & \mathbb{E}_\pi[\ell(f_\beta(\bar{X}))] \\
\text{s.t.} & \pi \in \mathcal{P}(\mathcal{X} \times \mathcal{X}) \\
& \mathbb{E}_\pi[c(\bar{X}, X)] \leq \rho, \ P_2\pi = \widehat{\mathbb{P}} \\
& \mathbb{E}_\pi[\bar{X}|X] = X \quad \widehat{\mathbb{P}}\text{-a.s.},
\end{cases}
\tag{3.1}
$$

and $\rho \geq 0$ is the radius of uncertainty set centered at $\widehat{\mathbb{P}}$. Note that because $\widehat{\mathbb{P}}$ is the empirical measure, the martingale constraint implies that the *conditional* expected value of the perturbation obtained by modifying each observed data point equals the observed data point itself. The quantity $L_\beta(\widehat{\mathbb{P}}, \rho)$ is referred to as the worst-case expected loss of the model parameter $\beta$ under the martingale DRO model. It is easy to see that $L_\beta(\widehat{\mathbb{P}}, \rho) \leq \mathbb{L}_\beta(\widehat{\mathbb{P}}, \rho)$, where $\mathbb{L}_\beta$ is defined as in (2.1). This is because the adversary in (3.1) has a smaller feasible set, and thus is less powerful than the adversary in (2.1). Hence, the martingale DRO solution for problem (3.1) is considered to be *less* conservative than the conventional DRO solution for problem (2.1).

The next result asserts that the martingale DRO problem coincides with the Tikhonov regularization problem under similar conditions of Proposition 2.2.

**Proposition 3.2** (Tikhonov equivalence). *Suppose that (i) the loss function $\ell(\cdot)$ is a convex quadratic function, i.e., $\nabla^2\ell(\cdot) = \gamma > 0$, and (ii) the feature mapping $f_\beta(\bar{X}) = \beta^\top \bar{X}$ is linear. Then we have*

$$
L_\beta(\widehat{\mathbb{P}}, \rho) = \mathbb{E}_{\widehat{\mathbb{P}}}[\ell(\beta^\top X)] + \frac{\gamma\rho}{2}\|\beta\|^2_{M^{-1}}.
$$

*If the Mahalanobis matrix $M$ is the identity matrix, the martingale DRO model (3.1) recovers the Tikhonov regularization problem.*

*Proof of Proposition 3.2.* By a change of the variable, let $\Delta = \bar{X} - X$ and we have

$$\sup_{\substack{\mathbb{E}_\pi[\|\Delta\|_M^2] \leq \rho \\ \mathbb{E}_\pi[\Delta|X]=0}} \mathbb{E}_\pi\left[\ell(\beta^\top(X+\Delta))\right] = \sup_{\substack{\mathbb{E}_\pi[\|\Delta\|_M^2] \leq \rho \\ \mathbb{E}_\pi[\Delta|X]=0}} \mathbb{E}_\pi\left[\ell(\beta^\top X) + \nabla\ell(\beta^\top X)\beta^\top\Delta + \frac{\gamma}{2}\|\beta^\top\Delta\|^2\right]$$

$$= \mathbb{E}_{\widehat{\mathbb{P}}}\left[\ell(\beta^\top X)\right] + \sup_{\substack{\mathbb{E}_\pi[\|\Delta\|_M^2] \leq \rho \\ \mathbb{E}_\pi[\Delta|X]=0}} \mathbb{E}_\pi\left[\frac{\gamma}{2}\|\beta^\top\Delta\|^2\right]$$

$$= \mathbb{E}_{\widehat{\mathbb{P}}}\left[\ell(\beta^\top X)\right] + \frac{\gamma\rho}{2}\|\beta\|_{M^{-1}}^2.$$

The last equality follows from the general Hölder's inequality. To achieve the equality, we can, for example, take a normally distributed random variable $C$ with mean 0 and variance $\rho$ and which is independent of $X$, and then let $\Delta = CM^{-1}\beta$. $\qquad\square$

**Example 3.3** (Linear regression). *Let $X^\top \triangleq (Y, Z^\top) \in \mathbb{R}^d$ and $\beta^\top \triangleq (1, -b^\top) \in \mathbb{R}^d$, we have $\beta^\top X = Y - b^\top Z$. For any $Q \in \mathbb{S}_{++}^{d-1}$, we take $M = \text{diag}(+\infty, Q)$, which implies that we do not allow transport of the response $Y$, then the problem (3.1) with $\gamma = 2$ becomes*

$$\min_b \left\{ \mathbb{E}_{\widehat{\mathbb{P}}}\left[(Y - b^\top Z)^2\right] + \rho\|b\|_{Q^{-1}}^2 \right\}.$$

In Appendix C.1, we give another instructive proof based on the strong duality result in Section 2. For general convex loss functions, we have the following certificate of robustness that provides upper and lower bound for the worst-case DRO loss in (3.1).

**Corollary 3.4** (General convex loss functions). *Suppose that (i) the loss function $\ell(\cdot)$ is $\mu$-strongly convex and $C$-smooth, that is, $\ell(\theta) \geq \ell(\theta') + \nabla\ell(\theta')(\theta'-\theta) + \frac{\mu}{2}(\theta'-\theta)^2$ and $\ell(\theta) \leq \ell(\theta') + \nabla\ell(\theta')(\theta'-\theta) + \frac{C}{2}(\theta'-\theta)^2$ hold for all $\theta, \theta' \in \mathbb{R}$, and (ii) the feature mapping $f_\beta(\bar{X}) = \beta^\top\bar{X}$ is linear. Then we have for any $\rho \geq 0$,*

$$\mathbb{E}_{\widehat{\mathbb{P}}}[\ell(f_\beta(X))] + \frac{\mu\rho}{2}\|\beta\|_{M^{-1}}^2 \leq L_\beta(\widehat{\mathbb{P}}, \rho) \leq \mathbb{E}_{\widehat{\mathbb{P}}}[\ell(f_\beta(X))] + \frac{C\rho}{2}\|\beta\|_{M^{-1}}^2.$$

**Example 3.5** (Logistic regression). *Let $X = YZ \in \mathbb{R}^d$, where $Z \in \mathbb{R}^d, Y \in \{\pm 1\}$, and $\ell(t) = \log(1 + \exp(-t))$, where $\ell(\cdot)$ satisfies Assumption (i) in Corollary 3.4 with $C = \frac{1}{4}$. Then we have*

$$L_\beta(\widehat{\mathbb{P}}, \rho) \leq \mathbb{E}_{\widehat{\mathbb{P}}}[\log(1 + \exp(-Y\beta^\top Z))] + \frac{\rho}{8}\|\beta\|_{M^{-1}}^2.$$

## 3.2 Optimal Transport-based DRO with Perturbed Martingale Constraints

Now we turn to the relaxation of the martingale constraint to improve upon Tikhonov regularization and gain more flexibility. We consider the perturbed martingale coupling based DRO model (perturbed martingale DRO):

$$\min_\beta \mathcal{L}_\beta(\widehat{\mathbb{P}}, \rho, \epsilon), \quad \text{where } \mathcal{L}_\beta(\widehat{\mathbb{P}}, \rho, \epsilon) \triangleq \begin{cases} \sup_\pi & \mathbb{E}_\pi[\ell(f_\beta(\bar{X}))] \\ \text{s.t.} & \pi \in \mathcal{P}(\mathcal{X} \times \mathcal{X}) \\ & \mathbb{E}_\pi[c(\bar{X}, X)] \leq \rho, \ P_2\pi = \widehat{\mathbb{P}} \\ & \|\mathbb{E}_\pi[\bar{X}|X] - X\|_M \leq \epsilon \quad \widehat{\mathbb{P}}\text{-a.s.} \end{cases} \tag{3.2}$$

The parameter $\epsilon$ controls the allowed violations of the martingale constraint for the adversary. It is trivial that if we set $\epsilon = 0$, then we obtain $\mathcal{L}_\beta(\widehat{\mathbb{P}}, \rho, 0) = L_\beta(\widehat{\mathbb{P}}, \rho)$ and we recover the exact martingale DRO model (3.1). If we set $\epsilon = +\infty$ then the martingale constraint becomes ineffective, thus we have $\mathcal{L}_\beta(\widehat{\mathbb{P}}, \rho, +\infty) = \mathbb{L}_\beta(\widehat{\mathbb{P}}, \rho)$ and model (3.2) collapses to the DRO formulation (2.1). We thus can think of $\epsilon$ as an interpolating parameter connecting two extremes: the conventional DRO model (2.1) (at $\epsilon = +\infty$) and the exact martingale DRO model (3.1) (at $\epsilon = 0$).

However, the resulting optimization problem (3.2) constitutes an infinite-dimensional optimization problem over probability distributions and thus appears to be computationally intractable. To overcome this issue, we leverage Theorem 2.3 and prove that the problem (3.2) is actually equivalent to a finite-dimensional problem. To begin with, we present one crucial proposition that can be applied to more general settings.

**Theorem 3.6** (General loss functions and feature mappings). *Suppose that the loss function $\ell(\cdot)$ and the feature mapping $f_\beta(\cdot)$ are upper semi-continuous. Then, for any $\rho > 0$ and $\epsilon > 0$, the perturbed martingale DRO model* (3.2) *admits*

$$\mathcal{L}_\beta(\widehat{\mathbb{P}}, \rho, \epsilon) = \inf_{\lambda \geq 0, \alpha} \lambda\rho + \frac{\epsilon}{N}\sum_{i=1}^{N}\|\alpha_i\|_{M^{-1}} + \frac{1}{N}\sum_{i=1}^{N}\sup_{\Delta_i}\left[\ell(f_\beta(X_i + \Delta_i)) - \alpha_i^\top\Delta_i - \lambda\|\Delta_i\|_M^2\right].$$
(3.3)

*Sketch of proof.* The key step is to decouple (3.2) as a two-layer optimization problem:

$$\mathcal{L}_\beta(\widehat{\mathbb{P}}, \rho, \epsilon) = \sup_{\|\eta_i\|_M \leq \epsilon \; \forall i} \quad \sup_\pi \quad \int_\mathcal{X}\ell(f_\beta(\bar{X}))\mathrm{d}\pi$$
$$\text{s.t} \quad \pi \in \mathcal{P}(\mathcal{X} \times \mathcal{X})$$
$$\int_{\mathcal{X}\times\mathcal{X}} c(\bar{X}, X)\mathrm{d}\pi \leq \rho, \; P_2\pi = \widehat{\mathbb{P}}$$
$$\int_{\mathcal{X}\times\mathcal{X}} \mathbb{I}_{X_i}(X) \cdot \bar{X}\mathrm{d}\pi = \frac{1}{N}(X_i + \eta_i) \quad \forall i \in [N].$$
(3.4)

Then we invoke Theorem 2.3 for the inner maximization problem over $\pi$ and apply the Sion's minimax theorem [28] for the outer maximization over $\eta$. The desired result is obtained. $\qquad\square$

Due to the specific structure of the quadratic cost and the linear feature mapping, the dual variable $\lambda$ admits a closed-form representation and $\alpha \in \mathbb{R}^{N \times d}$ in (3.3) can be reduced to $N$ parallel one-dimensional optimization problems. Upon this observation, we thus conduct an instructive and intuitive reformulation in Theorem 3.7 for linear regression with proof detailed in Appendix C.2.

**Theorem 3.7** (Linear regression). *Suppose that (i) the loss function $\ell(\cdot)$ is a convex quadratic function, i.e., $\nabla^2\ell(\cdot) = \gamma > 0$ and (ii) the feature mapping $f_\beta(\bar{X}) = \beta^\top\bar{X}$ is linear. Then, the perturbed martingale DRO model* (3.2) *admits:*

$$\mathcal{L}_\beta(\widehat{\mathbb{P}}, \rho, \epsilon) = \mathbb{E}_{\widehat{\mathbb{P}}}[\ell(f_\beta(X))] + \frac{\rho\|\beta\|_{M^{-1}}^2}{2} + \boxed{R(\beta)},$$
(3.5)

*where the additional regularizer is defined as $R(\beta) \triangleq \|\beta\|_{M^{-1}} \min_{s\in\mathbb{R}^N} \left(\frac{\epsilon}{N}\|s\|_1 + \sqrt{\frac{\rho}{N}}\|G_\beta - s\|_2\right)$ and $G_\beta = (\nabla\ell(\beta^\top X_1), \ldots, \nabla\ell(\beta^\top X_N))^\top \in \mathbb{R}^N$.*

Obviously, compared with the exact martingale constraint investigated in Proposition 3.2, the perturbed constraint we consider here involves an additional term $R(\beta)$. Fortunately, armed with Lemma D.3, we are able to shed light on its intuitive interpretation based on the quantitative relationship between modeling parameters $\epsilon$ and $\rho$. First, we discuss two extreme cases — there are interesting hidden connections between (3.2) and other existing regularization techniques.

**Remark 3.8.** *Intuitively, if $\epsilon$ is relatively small, the optimal $s^\star$ is zero which means that all perturbed martingale constraints will be active. Precisely, if $\epsilon^2 \leq \rho$, the additional regularization term satisfies $R(\beta) = \epsilon\mathbb{E}_{\widehat{\mathbb{P}}}[\|\nabla_X\ell(f_\beta(X))\|_{M^{-1}}]$, which is so-called Jacobian or input gradient regularizer in the literature when $M$ is an identity matrix. Recently, it has received much intention owing to its ability to improve adversarial robustness [32, 9]. Conversely, if $\rho$ is relatively small, the optimal $s^\star$ is equal to $G_\beta$, implying that none of perturbed martingale constraints is active. In fact, if $\epsilon^2 \geq N\rho$, we have $R(\beta) = \sqrt{\rho\mathbb{E}_{\widehat{\mathbb{P}}}[\|\nabla_X\ell(f_\beta(X))\|_{M^{-1}}^2]}$. Then, we can easily figure out that $\mathcal{L}_\beta(\widehat{\mathbb{P}}, \rho, \epsilon)$ can be reduced to the conventional OT-based DRO model in Proposition 2.2.*

Then for the middle case, it is natural to infer that only part of constraints will be active. In Lemma D.3, we justify this conjecture rigorously. We refer the reader to Appendix D for the details. As such, the proposed martingale DRO model takes the first step bridging the input gradient regularization and regularized square-root regression problem in a unified framework. On the other hand, it also opens up an exciting brand new avenue of robustified regularizers. In the next section, we validate its effectiveness for the adversarial training task.

# 4  Optimization Algorithms

To take advantage of the proposed perturbed martingale DRO model, a natural question here is whether we can address problem (3.2) in a tractable manner. In this section, we answer the above question in the affirmative by developing two different computational paradigms for linear regression and deep neural network respectively.

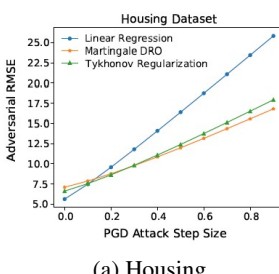
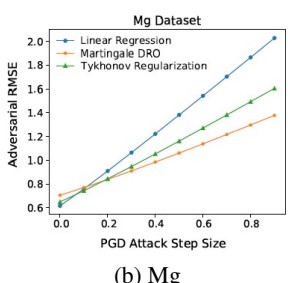
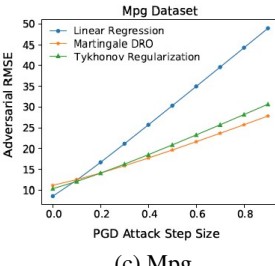

| (a) Housing | (b) Mg | (c) Mpg |

Figure 1: Compare the proposed martingale DRO with two standard benchmarks (linear regression and ridge regression) on three real world datasets — Housing, Mg and Mpg. The martingale DRO performs better than competing methods under large PGD step size.

## 4.1 Subgradient Method for Linear Regression

Unfortunately, the resulting formulation we conducted in Theorem 3.7 (i.e., (3.5)) for linear regression is potentially intractable for optimizing $(\beta, s)$ jointly. One obvious computational challenge here is that the overall problem is not necessarily convex in $(\beta, s)$, even if the original problem is convex over $\beta$. This is because the minima of convex functions is not convex. However, under a mild assumption, it is possible to give a reliable computational routine for solving the resulting problem over $\beta$ directly.

To start with, a key observation is $\mathcal{L}_\beta(\widehat{\mathbb{P}}, \rho, \epsilon)$ is convex over $\beta$. This is essentially from the fact that the pointwise supremum of a class of convex functions is still convex. To check the details, $\mathcal{L}_\beta(\widehat{\mathbb{P}}, \rho, \epsilon)$ is originally defined in (3.2) and $\mathbb{E}_\pi[\ell(f_\beta(\bar{X}))]$ is convex over $\beta$ for all $\pi$. Thus, a natural yet simple algorithm is subgradient method. The main difficulty is to obtain the correct subgradient oracle. Since the first two terms are smooth and strongly convex, we know that $R(\beta)$ is weakly convex and thus subdifferentially regular. All the subdifferential concepts are coincide. As such, we may simply use the Clarke subdifferential [6] in the sequel. Moreover, the sum rule is hold for computing the subgradient of $\mathcal{L}_\beta(\widehat{\mathbb{P}}, \rho, \epsilon)$ due to the weakly convexity. We have

$$\partial_\beta \mathcal{L}_\beta(\widehat{\mathbb{P}}, \rho, \epsilon) = \nabla_\beta \mathbb{E}_{\widehat{\mathbb{P}}}[\ell(f_\beta(X))] + \nabla_\beta \frac{\rho\|\beta\|_{M^{-1}}^2}{2} + \partial_\beta R(\beta).$$

The remaining question is how to compute the Clarke subdifferential of $R(\beta)$. When $\epsilon^2 \le \rho$ and $\epsilon^2 \ge N\rho$, $R(\beta)$ will enjoy the convex composite structure. Thus, we can get the correct subgradient by invoking the chain rule developed in [22, Theorem 10.6] directly. The more subtle and tricky case is the middle one — $\rho < \epsilon^2 < N\rho$. Without of loss generality, we assume that $M = I$ for simplicity. Based on [6, Theorem 2.3.9], we have

$$\partial_\beta R(\beta) \subseteq \text{Conv}\left\{ \partial_\beta \left( \frac{\epsilon}{N} \sum_{i=1}^N \|s_i^\star\beta\|_1 + \sqrt{\frac{\rho}{N} \sum_{i=1}^N \|(\nabla\ell(\beta^\top X_i) - s_i^\star)\beta\|_2^2} \right), \ s^\star \in \mathcal{S}(\beta) \right\},$$

where $\mathcal{S}(\beta)$ is the optimal solution set and $\text{Conv}\{\cdot\}$ denotes the convex hull. If we assume the inclusion here is tight, then the vanilla subgradient method will converge to the optimal solution with the rate $\mathcal{O}(1/\sqrt{K})$ [4]. Empirically, we find out that the resulting subgradient method works well and the violated case will never happen.

## 4.2 A New Principled Adversarial Training Procedure for Deep Learning

In this subsection, we develop inexact stochastic gradient-type methods for (3.3) and thus are able to realized the benefits of the proposed DRO model in adversarial learning tasks. As the nonconvexity of $f_\beta(\cdot)$, the inner maximization problem over $\Delta_i$ will be no longer tractable. Therefore, we leverage the methodology proposed in [27] to gain the computational efficiency, that is, regarding the dual variable as a modeling parameter. To proceed, we make the same smoothness assumption in [27, Assumption B] (i.e., see Assumption E.1 in appendix for details).

**Lemma 4.1** (Convex-concave minimax theorem without compactness). *Suppose that $f : \mathbb{R}^n \to \mathbb{R}$ is convex and level bounded and $g : \mathbb{R}^m \to \mathbb{R}$ is strongly convex. Then we have*

$$\min_{x\in\mathbb{R}^n} \max_{y\in\mathbb{R}^m} f(x) + x^\top Ay - g(y) = \max_{y\in\mathbb{R}^m} \min_{x\in\mathbb{R}^n} f(x) + x^\top Ay - g(y).$$

Leveraging Lemma 4.1 and the smoothness assumption, (3.3) leads to a simple and instructive form:

$$\min_{\beta} \frac{1}{N} \sum_{i=1}^{N} \max_{\|\Delta_i\|_M \leq \epsilon} \left[ \ell(f_{\beta}(X_i + \Delta_i)) - \lambda \|\Delta_i\|_M^2 \right]. \tag{4.1}$$

In contrast to the vanilla DRO model studied in [27], (4.1) further constrains the perturbation into a Euclidean ball with the correlation information $M$. Moreover, we can observe that the magnitude of $\epsilon$ decides how many martingale constraints will be active, which also perfectly matches our theoretical results and interpretations established for linear regression, see Theorem 3.7 and Remark 3.8.

From a computational viewpoint, if $\lambda$ is large enough (see Lemma E.2 for details), the inner maximization problem is strongly concave and thus the outer minimization problem over $\beta$ will be smooth. This motivates Algorithm 1, an inexact stochastic gradient method for Problem (4.1). The convergence guarantee is provided in [27, Theorem 2]. It is worthwhile mentioning that the resulting new principled adversarial training is extremely easy to implement by only adding three lines of *Pytorch* code based on [27]. We refer the interested readers to Appendix F for details.

---

**Algorithm 1:** Martingale Distributionally Robust Optimization with Adversarial Training

---

**Input :** Sampling distribution $\widehat{\mathbb{P}}$, stepsize sequence $\{t_k\}_{k=0}^{K-1}$;

**for** $k = 0, 1, 2, \cdots, K - 1$ **do**

> Sample $X^k \sim \widehat{\mathbb{P}}$ and find an $\eta$-approximate maximizer $\hat{\Delta}^k$ satisfying
>
> $$\|\hat{\Delta}^k - \Delta_k^\star\| \leq \eta, \quad \text{where} \quad \Delta_k^\star = \arg\max_{\|\Delta\|_M \leq \epsilon} \left\{ \ell(f_{\beta^k}(X^k + \Delta)) - \lambda \|\Delta\|_M^2 \right\}.$$
>
> Set $\beta^{k+1} \leftarrow \beta^k - t_k \nabla_\beta \ell(f_{\beta^k}(X^k + \hat{\Delta}^k))$.

---

## 5 Numerical Results

In this section, we validate the effectiveness of our methods (referred to as *martingale DRO*) on both linear regression and deep neural networks under the adversarial setting. All simulations are implemented using Python 3.8 on: (1) a computer running Windows 10 with a 2.80GHz, Intel(R) Core(TM) i7-1165G7 processor and 16 GB of RAM, and (2) Google Colab with NVIDIA Tesla P100 GPU and 16 GB of RAM. As for the adversarial setting, we consider three types of attack, the detailed definitions of which are collected in Appendix F.

### 5.1 Linear Regression

To start with, we demonstrate the effectiveness of the proposed martingale DRO model (3.5) with the quadratic loss function and linear feature mapping, i.e., $\ell(f_{\beta}(X)) = \frac{1}{2}(Y - b^\top Z)^2$ with $X^\top \triangleq (Y, Z^\top)$ and $\beta^\top \triangleq (1, -b^\top)$, where $Z$ is the feature vector and $Y$ is the target variable. In this experiment, we test our method on three LIBSVM regression real world datasets [1]. More specifically, we randomly select 60% of the data to train the models and the rest as our test data. To showcase the effectiveness of martingale DRO model under adversarial setting, we apply one-step projected-gradient method (PGM) attack [17] on test data and report the performance in terms of the the root-mean-square error (RMSE) on adversarial test data, where $\text{RMSE} \triangleq \sqrt{\frac{1}{N} \sum_i (\hat{\beta}^\top x_{\text{adv}}^{(i)} - y_{\text{adv}}^{(i)})^2}$ and $\hat{\beta}$ is the estimator of $\beta$. All numerical results with different step sizes for PGM attack are collected in Figure 1. As we mentioned in Proposition 3.2, since the exact martinagle DRO model (i.e., $\epsilon = 0$) is equivalent to Tikhonov regularization, we choose the same hyperparameter $\rho = 0.08$ for ridge regression and martingale DRO mode for fair comparison. We can observe that the martingale DRO can outperform other two benchmarks over three real world datasets consistently when the step size for the PGD attack is relatively large. This result also corroborates our theoretical intuition — the additional regularization $R(\beta)$ can further improve the adversarial robustness.

---

[1] https://www.csie.ntu.edu.tw/~cjlin/libsvmtools/datasets/regression.html

## 5.2 Deep Neural Network for Adversarial Training

We generate the synthetic training data $\{(Y_i, Z_i)\}_{i \in \mathcal{I}}$ with a wide margin as follows: generate i.i.d. $Z_j \sim N(0, I_2)$, where $Z \in \mathbb{R}^2$, $I_2$ is the identity matrix in $\mathbb{R}^2$; set $\mathcal{I} = \{j : \|Z_j\|_2 \notin (\sqrt{2}/\eta, \eta\sqrt{2})\}$, where $\eta = 1.6$; let $Y_i = \text{sign}(\|Z_i\|_2 - \sqrt{2})$, $\forall i \in \mathcal{I}$. We train a neural network with 3 hidden layers of size 4, 3 and 2 and ELU activations between layers. We compare our approach (cf. martingale DRO) with ERM and the conventional DRO approach developed in [27]. More details about the experiment setup are collected in Appendix F.

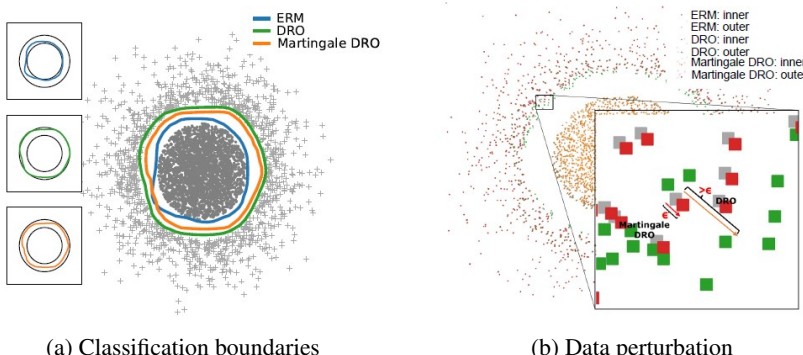

(a) Classification boundaries        (b) Data perturbation

Figure 2: Illustration of the performance comparison between competing methods. ERM tends to overfit to the inner class while DRO becomes too conservative due to unconstrained perturbation. The martingale DRO leaves bigger margins from the sample points than the other methods.

Figure 2 show the experimental results on the synthetic dataset. Test data are shown in darkgray and gray with different shapes, which are generated by the above-mentioned procedure with a smaller margin ($\eta = 1.2$). Classification boundaries are shown in blue, green, and orange for ERM, DRO, and martingale DRO respectively, as well as with the true class boundaries of the test data. Intuitively, the boundary generated by ERM is too close to the true inner boundary since the majority of points are of darkgray class, while the DRO approach pushes the classification boundary outwards. However, as illustrated in Figure 2(a), the DRO approach suffers from over-conservativeness and becomes entangled with the boundary of the outer gray class. In contrast, our martingale DRO boundary lands in between the two extremes and it leaves bigger margin. Figure 2(b) explicitly shows the qualitative difference between these two methods in terms of the perturbation to the data: the Martingale perturbation is constrained below $\epsilon$ while the DRO perturbation is unconstrained. Moreover, previously shown, decreasing non-zero $\epsilon$ pushes the perturbed martingale constraints towards the exact martingale constraints and forces the classification boundary increasingly inward. More results are in Appendix F.

Then, we validate our method on the MNIST dataset [15]. For the classifier, we train a neural network equipped with $8 \times 8$, $6 \times 6$, $5 \times 5$ convolutional filter layers and ELU activations followed by a fully connected layer and softmax output. To show the robustness of our method, we test the performance of four methods (ERM, DRO, Jacobian regularization [13] and martingale DRO) under the PGD and FGSM attacks (Definition F.1) with test error defined to be: $1 - \textit{classification accuracy}$.

In Figures 3a and 3b, our martingale DRO model outperforms the other methods and still provides robustness under the $\infty$-norm FGSM attacks. In Figure 3c, we show the performance of our model with different $\epsilon$. As expected, when $\epsilon$ is relatively small, the model is not flexible enough and shows large test error. Alternatively, as $\epsilon$ becomes large, our model will behave similarly to the original DRO model since the $\epsilon$-constraint is almost inactive in this case.

Figure 4 visualizes the different levels of robustness for the four methods. For each test data point, we perturb the image using the DRO attacks (Definition F.2) with decaying level of perturbation and respectively record the first perturbed images that each model correctly classifies. In Figure 4, the original label is 6 and all methods output the correct prediction, whereas in the adversarial example that the DRO model predicts 6, the correct classification seems unreasonable to human eyes (see Appendix F for more examples). This observation shows an insight that the original DRO model is too conservative in predicting and our model puts more constraints on the perturbation when training thus providing a model that is more consistent to human eyes.

| PGD Attack | ERM | DRO | Jacobian Regularization | Martingale DRO |
|---|---|---|---|---|
| $\epsilon = 0$ | 84.16% | 84.02% | 81.73% | 85.48% |
| $\epsilon = 0.04$ | 77.50% | 82.87% | 78.78% | 83.25% |
| $\epsilon = 0.08$ | 70.20% | 80.68% | 73.85% | 80.86% |

Table 1: Top-1 accuracy results with different levels of perturbation on CIFAR-10.

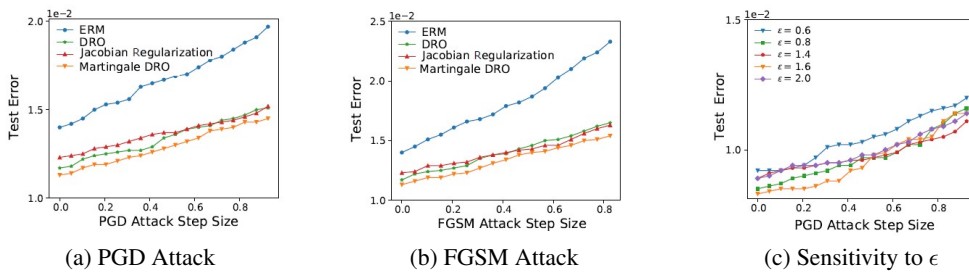

(a) PGD Attack      (b) FGSM Attack      (c) Sensitivity to $\epsilon$

Figure 3: Compare the proposed martingale DRO with ERM and DRO on the MNIST datasets under PGD and FGSM attack; compare the proposed martingale DRO with different values of $\epsilon$.

Experimental setup for CIFAR-10 [14]: For the classifier, we train a ResNet with the architecture in [12]. We optimize using Adam with a batch size of 128 for all methods. The learning rate starts from 0.01 and shrinks by $0.1^{\frac{\text{epoch}}{\text{total epochs}}}$, and each model is trained for 100 epochs. The simulations are implemented using Python 3.8 on Google Colab with TPU v2 and 16GB RAM. Similarly, we test the performance of four methods (ERM, DRO, Jacobian regularization and martingale DRO) under the PGD attack with different levels of perturbation, the results shown in Table 1 are consistent with those from the MNIST dataset.

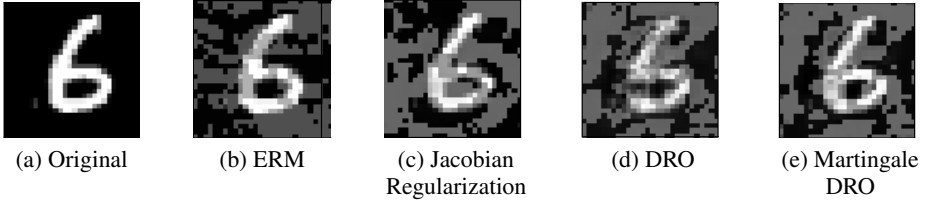

(a) Original      (b) ERM      (c) Jacobian Regularization      (d) DRO      (e) Martingale DRO

Figure 4: The largest DRO perturbation such that each model makes correct prediction.

## 6 Closing Remarks

In this paper, we find that the OT-based DRO model is equivalent to Tikhonov regularization when exact martingale constraints are imposed. Upon this interesting hidden connection, we introduce a new model called the perturbed martingale DRO model, which not only provides a unified viewpoint to several common robust methods but also leads to new regularization tools. Empirically, we validate the effectiveness of our model in addressing the conservativeness issue for the conventional DRO model. From the statistical perspective, how to optimally select the size of uncertainty regions and the perturbation size of the martingale constraint, is a natural problem to be further explored.

**Acknowledgements** Material in this paper is based upon work supported by the Air Force Office of Scientific Research under award number FA9550-20-1-0397. Additional support is gratefully acknowledged from NSF grants 1915967 and 2118199. Viet Anh Nguyen acknowledges the support from the CUHK's Improvement on Competitiveness in Hiring New Faculties Funding Scheme.

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
