## Broader impact

This work does not present any foreseeable societal consequence. While our contribution has a theoretical orientation, we believe that the structure of our method holds significant promise in the adversarial learning and robust optimization as we mentioned in the body context.

## A    Organization of the Appendix

We organize the appendix as follows:

- The proof details of Theorem 2.3 (cf. **Strong Duality Result**) is given in Section B.
- We collect all proof details of tractable reformulation results in Section C, including Proposition 3.2, Theorem 3.6 and Theorem 3.7.
- All useful technical lemmas are summarized in D.
- Convergence analysis of optimization algorithms are provided in Section E.
- Supplementary materials of numerical results are included in Section F.

## B    Strong Duality Result

To obtain the tractable reformulation result, we start to prove the strong duality theorem for a general class of martingale constraints-based Wasserstein DRO optimization problems (i.e., in the main context, we just provide the simplified version for simplicity):

$$
\begin{aligned}
\sup_{\mathbb{Q},\pi} \quad & \int_{\mathcal{X}} f(\bar{X}) \mathrm{d}\mathbb{Q} \\
\mathrm{s.\,t.} \quad & \mathbb{Q} \in \mathcal{P}(\mathcal{X}),\ \pi \in \mathcal{P}(\mathcal{X} \times \mathbb{X}) \\
& P_1\pi = \mathbb{Q},\, P_2\pi = \widehat{\mathbb{P}} \\
& \int_{\mathcal{X} \times \mathbb{X}} c(\bar{X}, X) \mathrm{d}\pi \leq \rho \\
& \mathbb{E}_\pi[\bar{X}|X] = \tilde{X} \quad \widehat{\mathbb{P}}\text{-a.s.}
\end{aligned}
\tag{Primal}
$$

Here,

- $f : \mathcal{X} \to \mathbb{R}$ is assumed to be upper semi-continuous and $\widehat{\mathbb{P}}$-integrable i.e., $f \in L^1(\widehat{\mathbb{P}})$.
- $\mathcal{P}(\mathcal{X})$ denotes the set of all Borel probability measures supported on $\mathcal{X}$.
- The cost function $c : \mathcal{X} \times \mathcal{X} \to [0, \infty]$ is a lower semicontinuous function satisfying $c(X, X) = 0$ for every $X \in \mathcal{X}$.
- $P_1\pi$ and $P_2\pi$ refer to the first and second marginal probability measure of $\pi$, that is, $(P_1\pi)(S) = \pi(S \times \mathcal{X})$ and $(P_2\pi)(S) = \pi(\mathcal{X} \times S)$ for any Borel subset $S$ of $\mathcal{X}$.
- For simplicity, let the reference measure be the empirical distribution $\widehat{\mathbb{P}} = \frac{1}{N} \sum_{i=1}^{N} \delta_{X_i}$ and $\mathbb{X} \triangleq \{X_1, X_2, \cdots, X_N\} \subset \mathcal{X}$.
- $\tilde{X}$ is the perturbed discrete distribution based on the empirical distribution $\widehat{\mathbb{P}}$ supported on $\{X_1 + \eta_1, \cdots, X_N + \eta_N\}$, i.e., $\tilde{\mathbb{P}} = \frac{1}{N} \sum_{i=1}^{N} \delta_{X_i + \eta_i}$.

The Lagrangian dual problem is derived as

$$
\min_{\substack{\lambda \in \mathbb{R}_+ \\ \alpha_i \in \mathbb{R}^d\ \forall i}} \lambda\rho + \sum_{i=1}^{N} \alpha_i^\top (X_i + \eta_i) + \frac{1}{N} \sum_{i=1}^{N} \max_{\bar{X}} \left[ f(\bar{X}) - \alpha_i^\top \bar{X} - \lambda c(\bar{X}, X_i) \right].
\tag{Dual}
$$

**Theorem B.1** (Restate Theorem 2.3 in a more general fashion)**.** *Suppose that (i) the reference measure $\widehat{\mathbb{P}}$ is the empirical distribution, i.e., $\widehat{\mathbb{P}} = \frac{1}{N} \sum_{i \in [N]} \delta_{X_i}$, (ii) $\tilde{X}$ follows from the perturbed empirical distribution, i.e., $\tilde{\mathbb{P}} = \frac{1}{N} \sum_{i \in [N]} \delta_{X_i + \eta_i}$ satisfying $X_i + \eta_i \in \mathrm{int}(\mathrm{cone}(\mathcal{X}))\ \forall i \in [N]$, and (iii) the ambiguity radius satisfies $\rho > 0$. Then strong duality holds, i.e., $\mathrm{Val}(Primal) = \mathrm{Val}(Dual)$.*

*Proof of Theorem B.1.* Since $\mathbb{Q} = P_1\pi$ a change of variables allows us to rewrite the objective function as

$$\int_{\mathcal{X}\times\mathbb{X}} f(\bar{X})\mathrm{d}\pi.$$

Then, as the reference measure $\widehat{\mathbb{P}} = \frac{1}{N}\sum_{i=1}^{N}\delta_{X_i}$, we can recast the marginal constraint $P_2\pi = \widehat{\mathbb{P}}$ as

$$\int_{\mathcal{X}\times\mathbb{X}} \mathbb{I}_{\mathcal{X}\times\{X_i\}}(\bar{X}, X)\mathrm{d}\pi = \frac{1}{N} \quad \forall i \in [N],$$

where $\mathbb{I}_{\mathcal{S}}$ is the indicator function of the set $\mathcal{S}$. Similarly, we can also reformulate the martingale constraint via further exploiting the discrete structure of the reference measure $\widehat{\mathbb{P}}$:

$$\int_{\mathcal{X}\times\mathbb{X}} \bar{X}\mathbb{I}_{X_i}(X)\mathrm{d}\pi = \frac{1}{N}\sum_{i=1}^{N}\int_{\mathcal{X}} \bar{X}\mathbb{I}_{X_i}(X_i)\mathbb{Q}^i(\mathrm{d}\bar{X})$$

$$= \frac{1}{N}\int_{\mathcal{X}} \mathbb{Q}^i(\mathrm{d}\bar{X}) = \frac{1}{N}(X_i + \eta_i).$$

Thus, we have

$$\int_{\mathcal{X}\times\mathbb{X}} \mathbb{I}_{X_i}(X)\cdot\bar{X}\mathrm{d}\pi = \frac{1}{N}(X_i + \eta_i) \quad \forall i \in [N],$$

where $X_i + \eta_i \in \mathcal{X}$. If we make the normalization of $\pi$ explicit, we obtain the following equivalent reformulation of Problem (Primal):

$$\begin{aligned}
\sup_{\pi\in M_+(\mathcal{X}\times\mathbb{X})} \quad & \int_{\mathcal{X}\times\mathbb{X}} f(\bar{X})\mathrm{d}\pi. \\
\text{s.t.} \quad & \int_{\mathcal{X}\times\mathbb{X}} \mathbb{I}_{\mathcal{X}\times\{X_i\}}(\bar{X}, X)\mathrm{d}\pi = \frac{1}{N} & \forall i \in [N] \\
& \int_{\mathcal{X}\times\mathbb{X}} \mathbb{I}_{X_i}(X)\cdot\bar{X}\mathrm{d}\pi = \frac{1}{N}(X_i + \eta_i) & \forall i \in [N] \\
& \int_{\mathcal{X}\times\mathbb{X}} c(\bar{X}, X)\mathrm{d}\pi \le \rho.
\end{aligned} \quad \text{(B.1)}$$

Here, $M_+(\mathcal{X}\times\mathbb{X})$ is the set of all non-negative Borel measures supported on $\mathcal{X}\times\mathbb{X}$ and the first integral constraint ensures that $\pi$ is a probability measure. As $\mathcal{M}_+(\mathcal{X}\times\mathbb{X})$ is a convex cone and all of constraints regarding $\pi$ are linear, problem (B.1) can be fitted into the standard primal problem in [26, (3.2)]. That is,

$$\begin{aligned}
\min_{\pi\in\mathcal{M}_+(\mathcal{X}\times\mathbb{X})} \quad & \langle f, \pi \rangle \\
\text{s.t.} \quad & \mathcal{A}(\pi) - b \in K,
\end{aligned} \quad \text{(B.2)}$$

where

$$K = \{0\}^{N+Nd} \times \mathbb{R}_{\le 0}, \quad b = \left(\frac{1}{N}\mathbf{e}_N, X_1 + \eta_1, \cdots, X_N + \eta_N, \rho\right),$$

and $\mathcal{A}$ is the linear mapping defined through the left hand side of the constraints in (B.1):

$$\mathcal{A}: \pi \mapsto \begin{bmatrix} \int_{\mathcal{X}\times\mathbb{X}} \mathbb{I}_{\mathcal{X}\times\{X_1\}}(\bar{X}, X)\mathrm{d}\pi \\ \vdots \\ \int_{\mathcal{X}\times\mathbb{X}} \mathbb{I}_{\mathcal{X}\times\{X_N\}}(\bar{X}, X)\mathrm{d}\pi \\ \int_{\mathcal{X}\times\mathbb{X}} \mathbb{I}_{X_1}(X)\cdot\bar{X}\mathrm{d}\pi \\ \vdots \\ \int_{\mathcal{X}\times\mathbb{X}} \mathbb{I}_{X_N}(X)\cdot\bar{X}\mathrm{d}\pi \\ \int_{\mathcal{X}\times\mathbb{X}} c(\bar{X}, X)\mathrm{d}\pi \end{bmatrix}$$

Next, we aim at invoking Proposition 3.4 in [26] to prove the strong duality. A sufficient condition is the generalized Slater condition, see (3.12) in [26]. That is, we have to check

$$b \in \mathrm{int}[\mathcal{A}(M_+(\mathcal{X}\times\mathbb{X})) - K], \quad \text{(B.3)}$$

where $\mathrm{int}(\cdot)$ is the interior of a set. As such,

$$\mathcal{A}(M_+(\mathcal{X}\times\mathbb{X})) = [0, +\infty]^N \times \mathrm{Range}(F)^N \times [0, \infty],$$

where $F: M_+(\mathcal{X}\times\mathbb{X}) \to \mathbb{R}^d$ satisfying $F(\pi) = \int_{\mathcal{X}\times\mathbb{X}} \mathbb{I}_{X_i}(X)\cdot\bar{X}\mathrm{d}\pi$. Then,

$$\mathcal{A}(M_+(\mathcal{X}\times\mathbb{X})) - K = [0, +\infty]^N \times \mathrm{Range}(F)^N \times [0, \infty].$$

To check the Slater condition, we validate each constraint separately.

- $\frac{1}{N} \in \mathrm{int}([0, +\infty])$, for all $i \in [N]$;

- As $X_i + \eta_i \in \mathrm{int}(\mathrm{cone}(\mathcal{X})), \forall i \in [N]$ and Lemma D.1 holds (i.e., $\mathrm{cone}(\mathcal{X}) \subseteq \mathrm{Range}(F)$), then $\frac{1}{N}(X_i + \eta_i) \in \mathrm{int}(\mathrm{Range}(F)), \forall i \in [N]$.

- Due to $\rho > 0$, we have $\rho \in \mathrm{int}([0, +\infty])$.

Then, we obtained the desirable result. At last, we derive the dual problem by the standard Lagrangian method following [26],

$$
\begin{aligned}
\mathcal{L}(\pi; \lambda, s, \alpha) =& \lambda\rho + \frac{1}{N}\sum_{i=1}^{N} s_i + \frac{1}{N}\sum_{i=1}^{N} \alpha_i^\top (X_i + \eta_i) \\
&+ \int_{\mathcal{X}\times\mathbb{X}} \left[ f(\bar{X}) - \sum_{i=1}^{N} \mathbb{I}_{X_i}(X)\cdot\alpha_i^\top\bar{X} - \lambda c(\bar{X}, X) - \sum_{i=1}^{N} s_i\mathbb{I}_{\mathcal{X}\times\{X_i\}}(\bar{X}, X) \right] \mathrm{d}\pi.
\end{aligned}
$$

Due to the strong duality result, we have

$$
\sup_{\pi\in M_+(\mathcal{X}\times\mathbb{X})} \min_{\lambda\geq 0, s, \alpha} \mathcal{L}(\pi; \lambda, s, \alpha) = \min_{\lambda\geq 0, s, \alpha} \sup_{\pi\in M_+(\mathcal{X}\times\mathbb{X})} \mathcal{L}(\pi; \lambda, s, \alpha).
$$

Moreover, since $\mathbb{X} = \{X_1, X_2, \cdots, X_N\}$, the nonnegative measure $\pi \in M_+(\mathcal{X}\times\mathbb{X})$ can be decomposed as $\pi(\bar{X}, X) = \sum_{i=1}^{N} w_i\mathbb{I}_{X_i}(X)\mathbb{Q}^i(\bar{X})$ where $w_i \geq 0, \forall i \in [N]$. Then, we have

$$
\begin{aligned}
&\min_{\lambda\geq 0, s, \alpha} \sup_{\pi\in M_+(\mathcal{X}\times\mathbb{X})} \mathcal{L}(\pi; \lambda, s, \alpha) \\
=& \min_{\lambda\geq 0, s, \alpha} \sup_{w_i\geq 0} \lambda\rho + \frac{1}{N}\sum_{i=1}^{N} s_i + \frac{1}{N}\sum_{i=1}^{N} \alpha_i^\top (X_i + \eta_i) \\
&+ \sum_{i=1}^{N} w_i \max_{\bar{X}} \left[ f(\bar{X}) - \alpha_i^\top\bar{X} - \lambda c(\bar{X}, X_i) - s_i \right] \\
=& \min_{\lambda\geq 0, s, \alpha} \sup_{w_i\geq 0} \lambda\rho + \sum_{i=1}^{N} \left(\frac{1}{N} - w_i\right) s_i + \frac{1}{N}\sum_{i=1}^{N} \alpha_i^\top (X_i + \eta_i) \\
&+ \sum_{i=1}^{N} w_i \max_{\bar{X}} \left[ f(\bar{X}) - \alpha_i^\top\bar{X} - \lambda c(\bar{X}, X_i) \right] \\
=& \min_{\lambda\geq 0, \alpha} \lambda\rho + \frac{1}{N}\sum_{i=1}^{N} \alpha_i^\top (X_i + \eta_i) + \frac{1}{N}\sum_{i=1}^{N} \max_{\bar{X}} \left[ f(\bar{X}) - \alpha_i^\top\bar{X} - \lambda c(\bar{X}, X_i) \right].
\end{aligned}
\tag{B.4}
$$

We complete the proof. □

## C   Proof Details of Tractable Reformulation Results

### C.1   Proof of Proposition 3.2

*Proof of Proposition 3.2.* Problem (3.1) can be recast into

$$
\begin{aligned}
\sup \quad & \int_{\mathcal{X}} \ell(\beta^\top\bar{X})\mathrm{d}\mathbb{Q} \\
\mathrm{s.\,t.} \quad & \mathbb{Q} \in \mathcal{P}(\mathcal{X}), \pi \in \mathcal{P}(\mathcal{X}\times\mathcal{X}) \\
& P_1\pi = \mathbb{Q}, P_2\pi = \widehat{\mathbb{P}} \\
& \int_{\mathcal{X}\times\mathbb{X}} c(X, \bar{X})\mathrm{d}\pi \leq \rho \\
& \int_{\mathcal{X}\times\mathbb{X}} \mathbb{I}_{X_i}(X)\cdot\bar{X}\mathrm{d}\pi = \frac{1}{N}X_i \quad \forall i \in [N].
\end{aligned}
\tag{C.1}
$$

Because $\widehat{\mathbb{P}}$ is the empirical measure and because any feasible measure $\pi$ satisfy the constraint $P_2\pi = \widehat{\mathbb{P}}$, the integral in the last two constraints of (C.1) is restricted to $\mathcal{X}\times\mathbb{X}$ (instead of $\mathcal{X}\times\mathcal{X}$) without any loss of optimality.

The trivial case $\rho = 0$ is easy to verify. To begin with, we focus on the case where $\rho > 0$. Here, we want to invoke the strong result, i.e., Theorem 2.3. Before getting into details, we check the conditions at first. $\ell(\cdot)$ is quadratic and thus upper semi-continuous; $\mathcal{X} = \mathbb{R}^d$ can help us to get rid of the mild regularity condition, that is, $X_i \in \text{int}(\mathbb{R}^d)$ automatically holds. Then, we get

$$L_\beta(\widehat{\mathbb{P}}, \rho) = \min_{\lambda \geq 0, \alpha} \lambda\rho + \frac{1}{N}\sum_{i=1}^N \alpha_i^\top X_i + \frac{1}{N}\sum_{i=1}^N \max_{\bar{X}} \left[\ell(\beta^\top \bar{X}) - \alpha_i^\top \bar{X} - \lambda c(\bar{X}, X_i)\right].$$

By a change of the variables, i.e., $\Delta_i = \bar{X} - X_i \; \forall i \in [N]$, we have

$L_\beta(\widehat{\mathbb{P}}, \rho)$

$$= \min_{\lambda \geq 0, \alpha} \lambda\rho + \frac{1}{N}\sum_{i=1}^N \max_{\Delta_i} \left[\ell(\beta^\top(X_i + \Delta_i)) - \alpha_i^\top \Delta_i - \lambda\|\Delta_i\|_M^2\right]$$

$$= \min_{\lambda \geq 0, \alpha} \lambda\rho + \frac{1}{N}\sum_{i=1}^N \max_{\Delta_i} \left[\ell(\beta^\top X_i) + \nabla\ell(\beta^\top X_i)\beta^\top \Delta_i + \frac{\gamma}{2}\|\beta^\top \Delta_i\|^2 - \alpha_i^\top \Delta_i - \lambda\|\Delta_i\|_M^2\right]$$

$$= \frac{1}{N}\sum_{i=1}^N \ell(\beta^\top X_i) + \min_{\lambda \geq 0, \alpha} \lambda\rho + \frac{1}{N}\sum_{i=1}^N \max_{\Delta_i} \left[(\nabla\ell(\beta^\top X_i)\beta - \alpha_i)^\top \Delta_i + \frac{\gamma}{2}\|\beta^\top \Delta_i\|^2 - \lambda\|\Delta_i\|_M^2\right].$$

Thus, the crux is the inner maximization problem. To proceed, we exhaust all possible cases. When $\lambda < \|\beta\|_{M^{-1}}^2\gamma/2$, it is easy to check that the inner maximization problem will go to $+\infty$ due to the general Cauchy-Schwarz inequality for the normed space $\|\beta^\top \Delta_i\|^2 \leq \|\beta\|_{M^{-1}}^2\|\Delta_i\|_M^2$. When $\lambda = \|\beta\|_{M^{-1}}^2\gamma/2$ and $\alpha_i \neq \nabla\ell(\beta^\top X_i)\beta$, the inner maximization problem will also go to $+\infty$. As such, we have

$$L_\beta(\widehat{\mathbb{P}}, \rho) \leq \frac{1}{N}\sum_{i=1}^N \ell(\beta^\top X_i) + \min_{\lambda \geq 0, \alpha} \lambda\rho + \frac{1}{N}\sum_{i=1}^N \max_{\Delta_i} \left[\frac{\gamma}{2}\|\beta^\top \Delta_i\|_M^2 - \lambda\|\Delta_i\|_M^2\right]$$

$$= \frac{1}{N}\sum_{i=1}^N \ell(\beta^\top X_i) + \min_{\lambda \geq 0, \alpha} \lambda\rho + \frac{1}{N}\sum_{i=1}^N \max_{\Delta_i} \left[\frac{\gamma}{2}\|\beta\|_{M^{-1}}^2\|\Delta_i\|_M^2 - \lambda\|\Delta_i\|_M^2\right]$$

$$= \frac{1}{N}\sum_{i=1}^N \ell(\beta^\top X_i) + \frac{\gamma\rho}{2}\|\beta\|_{M^{-1}}^2,$$

if $\lambda = \|\beta\|_{M^{-1}}^2\gamma/2$ and $\alpha_i = \nabla\ell(\beta^\top X_i)\beta$. At last, we focus on the left case and further prove the above inequality is the equality. If $\lambda > \|\beta\|_{M^{-1}}^2\gamma/2$, we have

$$\lambda\rho + \frac{1}{N}\sum_{i=1}^N \max_{\Delta_i} \left[(\nabla\ell(\beta^\top X_i)\beta - \alpha_i)^\top \Delta_i + \frac{\gamma}{2}\|\beta^\top \Delta_i\|^2 - \lambda\|\Delta_i\|_M^2\right] > \|\beta\|_{M^{-1}}^2\gamma/2 + 0.$$

The desirable result is obtained, that is,

$$L_\beta(\widehat{\mathbb{P}}, \rho) = \frac{1}{N}\sum_{i=1}^N \ell(\beta^\top X_i) + \frac{\gamma\rho}{2}\|\beta\|_{M^{-1}}^2 = \mathbb{E}_{\widehat{\mathbb{P}}}[\ell(\beta^\top X)] + \frac{\gamma\rho}{2}\|\beta\|_{M^{-1}}^2.$$

This completes the proof. $\qquad\square$

## C.2 Proof of Theorem 3.6 and Theorem 3.7

*Proof of Theorem 3.6.* To start with, we recast problem (3.2) into a two-layer optimization problem:

$$\mathcal{L}_\beta(\widehat{\mathbb{P}}, \rho, \epsilon) = \sup_{\|\eta_i\|_M \leq \epsilon \; \forall i} \; \sup_\pi \; \int_{\mathcal{X}} \ell(f_\beta(\bar{X}))\mathrm{d}\mathbb{Q}$$

$$\begin{aligned}
\text{s.t} \quad & \mathbb{Q} \in \mathcal{P}(\mathcal{X}), \pi \in \mathcal{P}(\mathcal{X} \times \mathcal{X}), \\
& P_1\pi = \mathbb{Q}, P_2\pi = \widehat{\mathbb{P}} \\
& \int_{\mathcal{X} \times \mathbb{X}} c(\bar{X}, X)\mathrm{d}\pi \leq \rho \\
& \int_{\mathcal{X} \times \mathbb{X}} \mathbb{I}_{X_i}(X) \cdot \bar{X}\mathrm{d}\pi = \frac{1}{N}(X_i + \eta_i) \quad \forall i \in [N],
\end{aligned} \tag{C.2}$$

where $\mathbb{X} = \{X_1, X_2, \cdots, X_N\}$. Then, we apply Theorem 2.3 (i.e., strong duality) to the inner maximization problem, i.e.,

$$\mathcal{L}_\beta(\widehat{\mathbb{P}}, \rho, \epsilon)$$

$$= \sup_{\|\eta_i\|_M \le \epsilon \; \forall i} \inf_{\lambda \ge 0, \alpha} \lambda\rho + \frac{1}{N} \sum_{i=1}^N \alpha_i^\top (X_i + \eta_i) + \frac{1}{N} \sum_{i=1}^N \sup_{\bar{X}_i} \left[ \ell(f_\beta(\bar{X}_i)) - \alpha_i^\top \bar{X}_i - \lambda c(\bar{X}_i, X_i) \right]$$

$$= \sup_{\|\eta_i\|_M \le \epsilon \; \forall i} \inf_{\lambda \ge 0, \alpha} \lambda\rho + \frac{1}{N} \sum_{i=1}^N \alpha_i^\top \eta_i + \frac{1}{N} \sum_{i=1}^N \sup_{\Delta_i} \left[ \ell(f_\beta(X_i + \Delta_i)) - \alpha_i^\top \Delta_i - \lambda\|\Delta_i\|_M^2 \right].$$

The second equality follows by setting $\bar{X}_i = X_i + \Delta_i$. As $0 < \epsilon < +\infty$ and $M$ is a positive definite matrix, the set $\{(\eta_1, \ldots, \eta_N) \; : \; \|\eta_i\|_M \le \epsilon \; \forall i \in [N]\}$ is a compact set. Consider the following mapping

$$(\eta, \lambda, \alpha) \mapsto \lambda\rho + \frac{1}{N} \sum_{i=1}^N \alpha_i^\top \eta_i + \frac{1}{N} \sum_{i=1}^N \sup_{\Delta_i} \left[ \ell(f_\beta(X_i + \Delta_i)) - \alpha_i^\top \Delta_i - \lambda\|\Delta_i\|_M^2 \right].$$

It is easy to see that this mapping is linear, and thus concave, in $\eta$. Moreover, it is convex in $(\lambda, \alpha)$ as the pointwise supremum of a class of convex functions (i.e., the inner function over $(\lambda, \alpha)$ is linear) is always convex. From Sion's minimax theorem [28], we can interchange the outer supremum and infimum operators to obtain

$$\mathcal{L}_\beta(\widehat{\mathbb{P}}, \rho, \epsilon)$$

$$= \inf_{\lambda \ge 0, \alpha} \sup_{\|\eta_i\|_M \le \epsilon \; \forall i} \lambda\rho + \frac{1}{N} \sum_{i=1}^N \alpha_i^\top \eta_i + \frac{1}{N} \sum_{i=1}^N \sup_{\Delta_i} \left[ \ell(f_\beta(X_i + \Delta_i)) - \alpha_i^\top \Delta_i - \lambda\|\Delta_i\|_M^2 \right].$$

For any feasible value of $(\lambda, \alpha)$, the optimal solution in $\eta_i$ is either

$$\eta_i^\star = M^{-1}\alpha_i \quad \text{or} \quad \eta_i^\star = -M^{-1}\alpha_i.$$

We thus have

$$\mathcal{L}_\beta(\widehat{\mathbb{P}}, \rho, \epsilon) = \inf_{\lambda \ge 0, \alpha} \lambda\rho + \frac{\epsilon}{N} \sum_{i=1}^N \|\alpha_i\|_{M^{-1}} + \frac{1}{N} \sum_{i=1}^N \sup_{\Delta_i} \left[ \ell(f_\beta(X_i + \Delta_i)) - \alpha_i^\top \Delta_i - \lambda\|\Delta_i\|_M^2 \right].$$

We complete the proof. $\qquad\square$

*Proof of Theorem 3.7.* Taking $\ell(f_\beta(X)) = \ell(\beta^\top X)$ with the second derivative of $\nabla^2 \ell(\cdot) = \gamma$ in (3.3), we have

$$\mathcal{L}_\beta(\widehat{\mathbb{P}}, \rho, \epsilon) = \inf_{\lambda \ge 0, \alpha} \lambda\rho + \frac{\epsilon}{N} \sum_{i=1}^N \|\alpha_i\|_{M^{-1}} + \frac{1}{N} \sum_{i=1}^N \sup_{\Delta_i} \left[ \ell(\beta^\top (X_i + \Delta_i)) - \alpha_i^\top \Delta_i - \lambda\|\Delta_i\|_M^2 \right]$$

$$= \inf_{\lambda \ge 0, \alpha} \lambda\rho + \frac{\epsilon}{N} \sum_{i=1}^N \|\alpha_i\|_{M^{-1}} +$$

$$\frac{1}{N} \sum_{i=1}^N \sup_{\Delta_i} \left[ \ell(\beta^\top X_i) + \nabla\ell(\beta^\top X_i)\beta^\top \Delta_i + \frac{\gamma}{2}\|\beta^\top \Delta_i\|^2 - \alpha_i^\top \Delta_i - \lambda\|\Delta_i\|_M^2 \right]$$

$$= \frac{1}{N} \sum_{i=1}^N \ell(\beta^\top X_i) + \inf_{\lambda \ge 0, \alpha} \lambda\rho + \frac{\epsilon}{N} \sum_{i=1}^N \|\alpha_i\|_{M^{-1}} +$$

$$\frac{1}{N} \sum_{i=1}^N \max_{\Delta_i} \left[ (\nabla\ell(\beta^\top X_i)\beta - \alpha_i)^\top \Delta_i + \frac{\gamma}{2}\|\beta^\top \Delta_i\|^2 - \lambda\|\Delta_i\|_M^2 \right]$$

$$= \frac{1}{N} \sum_{i=1}^N \ell(\beta^\top X_i) + \inf_{\lambda \ge 0, \alpha} \lambda\rho + \frac{\epsilon}{N} \sum_{i=1}^N \|\alpha_i\|_{M^{-1}} +$$

$$\frac{1}{N}\sum_{i=1}^{N}\sup_{\Delta_i}\left[(\nabla\ell(\beta^\top X_i)\beta - \alpha_i)^\top\Delta_i + \frac{\gamma}{2}\|\beta^\top\Delta_i\|^2 - \lambda\|\Delta_i\|_M^2\right].$$

Similar with the argument to proof Proposition 3.2 in the appendix, see section **??** for details, we can conclude that $0 \leq \lambda \leq \frac{\gamma}{2}\|\beta\|_{M^{-1}}^2$. As such, we analyze two cases separately.

**Case 1: suppose that the optimal value of $\lambda^\star = \frac{\gamma}{2}\|\beta\|_{M^{-1}}^2$.** As we discussed the exact martingale DRO mode in the last subsection , we have $\alpha_i^\star = \nabla\ell(\beta^\top X_i)\beta$ and

$$\mathcal{L}_1^\star(\widehat{\mathbb{P}}, \rho, \epsilon) = \frac{1}{N}\sum_{i=1}^{N}\ell(\beta^\top X_i) + \frac{\rho\gamma}{2}\|\beta\|_{M^{-1}}^2 + \frac{\epsilon}{N}\sum_{i=1}^{N}\|\nabla\ell(\beta^\top X_i)\beta\|_{M^{-1}}. \tag{C.3}$$

**Case 2: suppose that the optimal value of $\lambda^\star > \frac{\gamma}{2}\|\beta\|_{M^{-1}}^2$.** For any fixed $i = 1, \ldots, N$. Define

$$F(\lambda, \alpha) = \max_{\rho}\left[(\nabla\ell(\beta^\top X_i)\beta - \alpha_i)^\top\Delta_i + \frac{\gamma}{2}\|\beta^\top\Delta_i\|^2 - \lambda\|\Delta_i\|_M^2\right].$$

As $\lambda^\star > \frac{\gamma}{2}\|\beta\|_{M^{-1}}^2$, the inner maximization with respect to $\Delta_i$ is strongly convex. Consequently, it is necessary and sufficient to study its first-order optimality condition:

$$(\nabla\ell(\beta^\top X_i)\beta - \alpha_i) + (\gamma\beta\beta^\top - 2\lambda M)\Delta_i = 0. \tag{C.4}$$

Then, we obtain the optimal solution and the optimal value,

$$\Delta_i^\star = (2\lambda M - \gamma\beta\beta^\top)^{-1}(\nabla\ell(\beta^\top X_i)\beta - \alpha_i), \tag{C.5}$$

where the matrix inversion is valid as $\lambda^\star > \frac{\gamma}{2}\|\beta\|_{M^{-1}}^2$ and

$$\begin{aligned}
F(\lambda, \alpha) &= \lambda\|\Delta_i^\star\|_M^2 - \frac{\gamma}{2}\|\beta^\top\Delta_i^\star\|^2 \\
&= (\Delta_i^\star)^\top\left(\lambda M - \frac{\gamma}{2}\beta\beta^\top\right)^{-1}\Delta_i^\star \\
&= \frac{1}{4}(\nabla\ell(\beta^\top X_i)\beta - \alpha_i)^\top\left(\lambda M - \frac{\gamma}{2}\beta\beta^\top\right)^{-1}(\nabla\ell(\beta^\top X_i)\beta - \alpha_i).
\end{aligned}$$

For simplicity, let us ignore the empirical loss at first, which is the constant w.r.t. the dual variables $\lambda$ and $\alpha$.

$$\min_{\lambda, \alpha}\lambda\rho + \frac{\epsilon}{N}\sum_{i=1}^{N}\|\alpha_i\|_{M^{-1}} + \frac{1}{N}\sum_{i=1}^{N}\max_{\Delta_i}\left[(\nabla\ell(\beta^\top X_i)\beta - \alpha_i)^\top\Delta_i + \frac{\gamma}{2}\|\beta^\top\Delta_i\|^2 - \lambda\|\Delta_i\|^2\right]$$

$$= \min_{\lambda > \frac{\gamma}{2}\|\beta\|_{M^{-1}}^2, \alpha}\lambda\rho + \frac{\epsilon}{N}\sum_{i=1}^{N}\|\alpha_i\|_{M^{-1}} +$$

$$\frac{1}{4N}\sum_{i=1}^{N}(\nabla\ell(\beta^\top X_i)\beta - \alpha_i)^\top\left(\lambda M - \frac{\gamma}{2}\beta\beta^\top\right)^{-1}(\nabla\ell(\beta^\top X_i)\beta - \alpha_i).$$

The resulting structure of $(\lambda, \alpha)$ is still quite complicated. To further characterize the structure of the optimal solution, we utilize the parallel structure of $\alpha$ and focus on the corresponding subproblem as follow:

$$\min_{\alpha_i}\epsilon\|\alpha_i\| + \frac{1}{4}(\nabla\ell(\beta^\top X_i)\beta - \alpha_i)^\top\left(\lambda I - \frac{\gamma}{2}\beta\beta^\top\right)^{-1}(\nabla\ell(\beta^\top X_i)\beta - \alpha_i). \tag{C.6}$$

By the Sherman–Morrison Formula (i.e., see Fact D.2), we have

$$\begin{aligned}
\left(\lambda M - \frac{\gamma}{2}\beta\beta^\top\right)^{-1} &= (\lambda M)^{-1} + \frac{(\lambda M)^{-1}(\frac{\gamma}{2}\beta\beta^\top)(\lambda M)^{-1}}{1 - \frac{\gamma}{2}\beta^\top(\lambda M)^{-1}\beta} \\
&= \frac{1}{\lambda}M^{-1} + \frac{\gamma M^{-1}\beta\beta^\top M^{-1}}{\lambda(2\lambda - \gamma\|\beta\|_{M^{-1}}^2)}.
\end{aligned}$$

Together with (C.6),

$$\min_{\alpha_i} \epsilon \|\alpha_i\|_{M^{-1}} + \frac{1}{4} (\nabla\ell(\beta^\top X_i)\beta - \alpha_i)^\top \left( \frac{1}{\lambda} M^{-1} + \frac{\gamma M^{-1}\beta\beta^\top M^{-1}}{\lambda(2\lambda - \gamma\|\beta\|_{M^{-1}}^2)} \right) (\nabla\ell(\beta^\top X_i)\beta - \alpha_i).$$

Similarly, as the minimization problem w.r.t. $\beta$ is strongly convex, it is sufficient to study its first-order optimality condition. WLOG, we can assume the optimal solution $\alpha_i \neq 0$ to get rid of the non-smooth point. Then, we have

$$0 = \frac{\epsilon M^{-1}\alpha_i}{\|\alpha_i\|_{M^{-1}}} + \frac{1}{2} \left( \frac{1}{\lambda} M^{-1} + \frac{\gamma M^{-1}\beta\beta^\top M^{-1}}{\lambda(2\lambda - \gamma\|\beta\|_{M^{-1}}^2)} \right) (\nabla\ell(\beta^\top X_i)\beta - \alpha_i)$$

$$= \left( \frac{\epsilon}{\|\alpha_i\|_{M^{-1}}} - \frac{1}{2\lambda} \right) M^{-1}\alpha_i + \left( \frac{\nabla\ell(\beta^\top X_i)}{2\lambda} + \frac{\gamma \left(\nabla\ell(\beta^\top X_i)\|\beta\|_{M^{-1}}^2 - \beta^\top M^{-1}\alpha_i\right)}{\lambda(2\lambda - \gamma\|\beta\|_{M^{-1}}^2)} \right) M^{-1}\beta.$$

It is easy to observe that the optimal solution $\alpha_i$ is parallel to $\beta$. The conclusion is also valid for the corner case $\alpha_i = 0$. Consequently, problem (C.6) can be reduced to a one-dimensional problem, i.e., $\alpha_i = s_i\beta$,

$$\min_{s_i} \epsilon\|\beta\|_{M^{-1}}|s_i| + \frac{1}{4}\left( \frac{\|\beta\|_{M^{-1}}^2}{\lambda} + \frac{\gamma\|\beta\|_{M^{-1}}^4}{\lambda(2\lambda - \gamma\|\beta\|_{M^{-1}}^2)} \right)(\nabla\ell(\beta^\top X_i) - s_i)^2. \qquad (C.7)$$

Putting all pieces together, we get

$$\min_{\lambda > \frac{\gamma}{2}\|\beta\|_{M^{-1}}^2, s} \lambda\rho + \frac{\epsilon\|\beta\|_{M^{-1}}}{N}\|s\|_1 + \frac{1}{4N}\left( \frac{\|\beta\|_{M^{-1}}^2}{\lambda} + \frac{\gamma\|\beta\|_{M^{-1}}^4}{\lambda(2\lambda - \gamma\|\beta\|_{M^{-1}}^2)} \right) \sum_{i=1}^N (\nabla\ell(\beta^\top X_i) - s_i)^2$$

$$= \min_{\lambda > \frac{\gamma}{2}\|\beta\|_{M^{-1}}^2, s} \lambda\rho + \frac{\epsilon\|\beta\|_{M^{-1}}}{N}\|s\|_1 + \frac{1}{4N}\left( \frac{\|\beta\|_{M^{-1}}^2}{\lambda} + \frac{\gamma\|\beta\|_{M^{-1}}^4}{\lambda(2\lambda - \gamma\|\beta\|_{M^{-1}}^2)} \right) \|G_\beta - s\|_2^2$$

$$= \min_{\lambda > \gamma/2, s} \lambda\rho\|\beta\|_{M^{-1}}^2 + \frac{\epsilon\|\beta\|_{M^{-1}}}{N}\|s\|_1 + \frac{1}{4N}\left( \frac{1}{\lambda} + \frac{\gamma}{\lambda(2\lambda - \gamma)} \right) \|G_\beta - s\|_2^2,$$

$$= \min_{\lambda > \gamma/2, s} \lambda\rho\|\beta\|^2 + \frac{\epsilon\|\beta\|}{N}\|s\|_1 + \frac{2}{4N(2\lambda - \gamma)}\|G_\beta - s\|_2^2,$$

where $G_\beta = (\nabla\ell(\beta^\top X_1), \cdots, \nabla\ell(\beta^\top X_N))$. By changing the variables $\theta = \frac{2}{2\lambda - \gamma}$ and $\lambda = \frac{1}{\theta} + \frac{\gamma}{2}$ where $\theta > 0$,

$$\min_s \frac{\gamma\rho}{2}\|\beta\|_{M^{-1}}^2 + \frac{\epsilon}{N}\|\beta\|_{M^{-1}}\|s\|_1 + \min_{\theta > 0} \frac{\rho}{\theta}\|\beta\|_{M^{-1}}^2 + \frac{\theta}{4N}\|G_\beta - s\|_2^2.$$

Here, the optimal solution $\theta^\star$ is

$$\theta^\star = \frac{2\sqrt{N\rho}\|\beta\|_{M^{-1}}}{\|G_\beta - s\|}.$$

Consequently, we have

$$\mathcal{L}_2^\star(\widehat{\mathbb{P}}, \rho, \epsilon) = \mathbb{E}_{\widehat{\mathbb{P}}}[\ell(\beta^\top X)] + \frac{\gamma\rho}{2}\|\beta\|_{M^{-1}}^2 + \|\beta\|_{M^{-1}} \min_s \left( \frac{\epsilon}{N}\|s\|_1 + \sqrt{\frac{\rho}{N}}\|G_\beta - s\|_2 \right). \quad (C.8)$$

By applying Lemma D.3, we have

$$\min_s \left( \frac{\epsilon}{N}\|s\|_1 + \sqrt{\frac{\rho}{N}}\|G_\beta - s\|_2 \right) = \frac{\epsilon}{N}\|G_\beta\|_1$$

when $\epsilon \leq \sqrt{\rho}$. Then, combining these two cases, we can obtain,

$$\mathcal{L}_\beta(\widehat{\mathbb{P}}, \rho, \epsilon) = \mathbb{E}_{\widehat{\mathbb{P}}}[\ell(\beta^\top X)] + \frac{\gamma\rho}{2}\|\beta\|_{M^{-1}}^2 + \|\beta\|_{M^{-1}} \min_s \left( \frac{\epsilon}{N}\|s\|_1 + \sqrt{\frac{\rho}{N}}\|G_\beta - s\|_2 \right).$$

We complete the proof. $\qquad\qquad\qquad\qquad\qquad\qquad\qquad\qquad\qquad\qquad\qquad\qquad\qquad\qquad\qquad\qquad\qquad\square$

# D   Useful Technical Lemmas

**Lemma D.1.** *Suppose that $F : \mathcal{M}_+(A) \to \mathbb{R}^d$ defined by $F(\mu) = \int_A X\mu(\mathrm{d}x)$, then we have $\mathrm{cone}(A) \subseteq \mathrm{Range}(F)$.*

*Proof of Lemma D.1.* Recall that

$$\mathrm{cone}(A) = \left\{ \sum_{i=1}^{k} w_i x_i \; : \; x_i \in A, \; w_i \in \mathbb{R}_{\geq 0}, \; k \in \mathbb{N} \right\}. \tag{D.1}$$

For any $x \in \mathrm{cone}(A)$, then there exists $x_1, x_2, \ldots, x_k \in A$ and $\{w_i\}_{i=1}^{k} \geq 0$ such that $x = \sum_{i=1}^{k} w_i x_i$. Pick $\mu = \sum_{i=1}^{k} w_i \delta_{x_i}$, where $\delta_{x_i}$ are Dirac's delta measure at $x_i$. Then $\mu \in \mathcal{M}_+(A)$, and

$$x = \sum_{i=1}^{k} w_i x_i = \sum_{i=1}^{k} w_i \int_A x\delta_{x_i}(\mathrm{d}x) = \int_A x\mu(\mathrm{d}x),$$

which leads to the postulated claim. $\qquad\square$

**Fact D.2** (Sherman–Morrison Formula). *Suppose $A \in \mathbb{R}^{n \times n}$ is an invertible square matrix and $u, v \in \mathbb{R}^n$ are column vectors. Then $A + uv^\top$ is invertible if and only if $1 + v^\top A^{-1}u \neq 0$. In this case,*

$$\left(A + uv^\top\right)^{-1} = A^{-1} - \frac{A^{-1}uv^\top A^{-1}}{1 + v^\top A^{-1}u}.$$

*Here, $uv^\top$ is the outer product of two vectors $u$ and $v$.*

**Lemma D.3.** *Suppose that $y \in \mathbb{R}^d$ satisfying $|y_1| \leq |y_2| \leq \cdots \leq |y_d|$ and $\vartheta > 0$. Then, there exist $1 < j < d$ and $\alpha > 0$ such that the problem*

$$\min_{x \in \mathbb{R}^d} \; \|x\|_1 + \vartheta\|y - x\|_2$$

*admits the following optimal solution*

$$x^\star(\vartheta) = \begin{cases} \mathbf{0} & \text{if } \vartheta \leq \frac{\|y\|_2}{\|y\|_\infty}, \\ [\mathbf{0}_{1:j}, y_{j+1:d} - \alpha\,\mathrm{sign}(y_{j+1:d})] & \text{if } \frac{\|y\|_2}{\|y\|_\infty} < \vartheta < \sqrt{d}, \\ y & \text{if } \vartheta \geq \sqrt{d}. \end{cases} \tag{D.2}$$

*Proof of Lemma D.3.* The basic strategy here is to check the first-order optimality condition.

- If $\vartheta \leq \frac{\|y\|_2}{\|y\|_\infty}$ we have

$$0 \in \partial\|x\|_1|_{x=0} - \vartheta\frac{y}{\|y\|}$$

  holds. Thus, 0 is the optimal solution.

- Moreover, if $\vartheta \geq \sqrt{d}$, we have

$$0 \in \mathrm{sign}(y) + \vartheta\partial\|x - y\|_2|_{x=y}$$

  holds as $v \in \partial\|x - y\|_2|_{x=y}$ satisfies $\|v\|_2 \leq 1$.

- The most complicated case is the middle one, i.e., $\frac{\|y\|_2}{\|y\|_\infty} < \vartheta < \sqrt{d}$. Here, we are trying to characterize the structure of the optimal solution. Without of loss generality, we assume that $y = \mathrm{sort}(y,'\,\mathrm{abs}')$, i.e., sorted by its absolute value. Still, we focus on its first-order optimality condition:

$$0 \in \partial\|x\|_1 + \vartheta\frac{x - y}{\|x - y\|_2},$$

  as $x$ cannot equal to $y$ derived from the condition $\frac{\|y\|_2}{\|y\|_\infty} < \vartheta < \sqrt{d}$. Furthermore, the optimal solution $x$ shares the same sign of $y$ and $|x_i| \leq |y_i|, \forall i \in [d]$, otherwise you can always decrease the objective value by changing the sign. Next, we will argue that the

optimal solution admits $x_i^* = 0$ for some index $i$. We prove it by contradiction. If we assume $x^* \neq 0$, there exists a constant $\alpha > 0$ such that

$$x^* = y - \alpha \operatorname{sign}(y).$$

Then, the first-order optimality condition will not hold, i.e.,

$$\operatorname{sign}(y) + \vartheta \frac{\alpha \operatorname{sign}(y)}{\|\alpha \operatorname{sign}(y)\|_2} \neq 0,$$

as $\vartheta < \sqrt{d}$. As such, there exist $1 < j < d$ and a constant $\alpha > 0$ such that $x^* = [\mathbf{0}_{1:j}, y_{j+1:d} - \alpha \operatorname{sign}(y_{j+1:d})]$.

$\square$

# E   Convergence Analysis of Optimization Algorithms

Denote $\ell(f_\beta(X)) = h(\beta, X)$ and we make the following blanket assumption:

**Assumption E.1.** *The loss function* $h : \Omega \times \mathcal{X} \to \mathbb{R}$ *satisfies the Lipschitzian smoothness conditions*

$$\begin{aligned}
\|\nabla_\beta h(\beta_1, X) - \nabla_\beta h(\beta_2, X)\| &\leq C_{\beta\beta} \|\beta_1 - \beta_2\|, \\
\|\nabla_X h(\beta, X_1) - \nabla_X h(\beta, X_2)\| &\leq C_{XX} \|X_1 - X_2\|, \\
\|\nabla_\beta h(\beta, X_1) - \nabla_\beta h(\beta, X_2)\| &\leq C_{\beta X} \|X_1 - X_2\|, \\
\|\nabla_X h(\beta_1, X) - \nabla_X h(\beta_2, X)\| &\leq C_{X\beta} \|\beta_1 - \beta_2\|,
\end{aligned}$$

*where* $\Omega \subset \mathbb{R}^d$ *is a closed convex set.*

**Derivation of** (4.1)

$$\min_\beta \frac{1}{N} \sum_{i=1}^N \min_{\alpha_i} \max_{\Delta_i} \left[ \ell(f_\beta(X_i + \Delta_i)) - \alpha_i^\top \Delta_i - \lambda \|\Delta_i\|_M^2 + \epsilon \|\alpha_i\|_{M^{-1}} \right]$$

$$\overset{(a)}{=} \min_\beta \frac{1}{N} \sum_{i=1}^N \max_{\Delta_i} \min_{\alpha_i} \left[ \epsilon \|\alpha_i\|_{M^{-1}} - \alpha_i^\top \Delta_i + \ell(f_\beta(X_i + \Delta_i)) - \lambda \|\Delta_i\|_M^2 \right] \qquad \text{(E.1)}$$

$$= \min_\beta \frac{1}{N} \sum_{i=1}^N \max_{\|\Delta_i\|_M \leq \epsilon} \left[ \ell(f_\beta(X_i + \Delta_i)) - \lambda \|\Delta_i\|_{M^{-1}}^2 \right],$$

where equality $(a)$ follows from the following minimax theorem as the inner maximization over $\Delta_i$ is strongly concave and $\|\alpha_i\|_{M^{-1}}$ is level bounded.

*Proof of Lemma 4.1.* By invoking the general best-case primal-dual relations given in [22, Corollary 11.40 (d)], the key ingredient is to check the boundedness of

$$\left\{ x \in \mathbb{R}^n : x = \arg\min_x \max_y \left\{ f(x) + x^\top A y - g(y) \right\} \right\}$$

and

$$\left\{ y \in \mathbb{R}^m : y = \arg\max_y \min_x \left\{ f(x) + x^\top A y - g(y) \right\} \right\}.$$

For the purpose of this proof, we use $f^*$ to denote the convex conjugate of $f$, formally defined as

$$f^*(z) = \max_{x \in \mathbb{R}^n} z^\top x - f(x).$$

Similarly, $g^*$ is the conjugate of $g$. We have:

- As $f(x)$ is level-bounded, $\max_y f(x) + x^\top A y - g(y) = f(x) + g^*(A^\top x)$ is also level bounded.

- As $g(y)$ is strongly convex, $\min_x f(x) + x^\top A y - g(y) = -f^*(-Ay) - g(y)$ is strongly concave and thus its optimal solution set is compact.

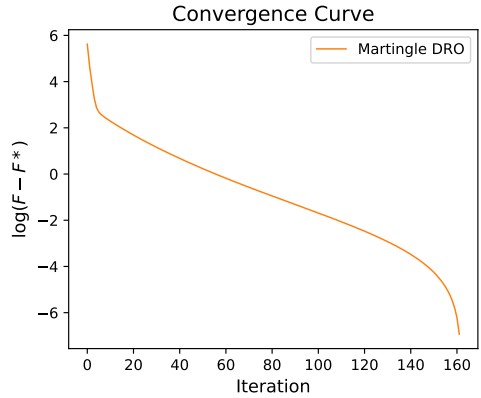

The proof is complete. □

**Lemma E.2.** *Let $h : \Omega \times \mathcal{X} \to \mathbb{R}$ be differentiable and $\phi_\lambda(\beta, X) = \sup_{\|\Delta\|_M \leq \epsilon}\{h(\beta, X + \Delta) - \lambda\|\Delta\|_M^2\}$. Suppose that Assumption E.1 holds and $\lambda > \sigma_{\min}(M)C_{XX}$, where $\sigma_{\min}(M)$ is the minimum eigenvalue of $M$. Then, $\phi_\lambda(\cdot, X)$ is differentiable.*

*Proof.* As the set $\|\Delta\|_M \leq \epsilon$ is a compact set whenever $0 < \epsilon < \infty$, we know the function $\phi_\lambda(\beta, X)$ is subsmooth function, see Definition 10.29 in [22]. Furthermore, since $h(\beta, \cdot)$ is $L$-smooth and $\lambda > \sigma_{\min}(M)C_{XX}$, we know that $h(\beta, X + \Delta) - \lambda\|\Delta\|_M^2$ is $(\lambda - \sigma_{\min}(M)C_{XX})$-strongly concave with respect to $\Delta$. Thus, the inner maximizer is unique and we can invoke [22, Theomrem 10.31] (i.e., an extension of envelope theorem) to obtain the differentiablity.

Compared with Lemma 1 in [27], our proof here is simpler as we utilize the compactness condition. □

Recall that

$$\min_\beta F(\beta) := \frac{1}{N}\sum_{i=1}^{N}\underbrace{\max_{\|\Delta_i\|_M \leq \epsilon}\left[\ell(f_\beta(X_i + \Delta_i))) - \lambda\|\Delta_i\|_M^2\right]}_{\phi_\lambda(\beta, X_i)}. \tag{E.2}$$

**Theorem E.3** (Convergence of nonconvex SGD; Adopted from Theorem 2 in [27]). *Suppose that $\Delta_F \geq F(\beta^0) - \inf_\beta F(\beta)$ and $\mathbb{E}\left[\|\nabla F(\beta) - \nabla_\beta\phi_\gamma(\beta, X)\|_2^2\right] \leq \sigma^2$ and we take constant stepsizes $\alpha = \sqrt{\frac{\Delta_F}{L_\phi K \sigma^2}}$ where $L_\phi := C_{\beta\beta} + \frac{C_{\beta X}C_{X\beta}}{\lambda - \sigma_{\min}(M)C_{XX}}$. For $K \geq \frac{L_\phi \Delta_F}{\sigma^2}$, Algorithm 1 satisfies*

$$\frac{1}{K}\sum_{k=0}^{K-1}\mathbb{E}\left[\|\nabla F(\beta^k)\|_2^2\right] - \frac{4C_{\beta X}^2}{\lambda - \sigma_{\min}(M)C_{XX}}\epsilon \leq 4\sigma\sqrt{\frac{L_\phi \Delta_F}{K}}.$$

## F  Supplementary Experiments

First we introduce the attack methods we use in the experiments of adversarial training.

**Definition F.1** (PGD/FGSM attack). *For any model parameter $\beta$, let*

$$\Delta z_i(\beta) \triangleq \underset{\|\eta\|_p \leq \xi}{\arg\max}\left\{\nabla_z \ell(f_\beta(z_i, y_i))^\top \eta\right\} \quad and \quad \tilde{z}_i \triangleq \Pi_{\mathcal{B}_{\xi,p}(z_i)}\left\{z_i + \alpha\,\Delta z_i(\beta)\right\},$$

*where $\xi$ is the attack step size, $\alpha$ is a pre-specified hyperparameter and $\Pi$ denotes the projection onto $\mathcal{B}_{\xi,p}(z_i) \triangleq \{z : \|z - z_i\|_p \leq \xi\}$. When $p = \infty$, the attack is reduced to the Fast Gradient Sign Method (FGSM) [11]. As in [27], we also consider the Euclidean case $p = 2$, which is a general version of Projected Gradient Descent (PGD) [17] with one step.*

**Definition F.2** (DRO attack). *For any parameter $\beta$, let*

$$\bar{z}_i \triangleq \underset{z \in \mathbb{R}^d}{\arg\max} \left\{ \ell(f_\beta(z_i, y_i)) - \gamma \|z - z_i\|_2^2 \right\},$$

*where $\gamma$ is is a pre-specified hyperparameter.*

As for the setup of the experiments on the synthetic data (i.e., Figure 2), we use $\lambda = 2$ for both the DRO approach and our approach. Further in our approach, we use a sequence of $\epsilon \in \{0.2, 0.22, \ldots, 1.5\}$ to demonstrate the effectiveness of our approach for being less conservative then the conventional DRO approach. Full results are shown in Figure 5.

As for the setup of the experiments on the MNIST dataset (Figures 3, 4), $\mathbb{E}_{\widehat{\mathbb{P}}}\|X\|_2 = 9.21$ and we choose $\lambda = 0.04\mathbb{E}_{\widehat{\mathbb{P}}}\|X\|_2$ for training the original DRO and Martingale DRO model. Additionally, we choose $\epsilon = 1.2$ in our model, which is smaller than the average $L^2$ norm of the perturbations suggested by the original DRO model when training on the MNIST dataset. As for the DRO attack, we choose an increasing sequence of $\gamma$ (corresponding to a decaying sequence of perturbation) and collect the images after the largest perturbation so that these methods can classify correctly. Full results are shown in Figures 6, 7.

All the hyperparameters conducted in this section have been fine-tuned via grid search for optimal performance.

In the following, we want to highlight that our method can be applied within three lines of *PyTorch* code modification based on the original DRO approach, which is due to the formulation (4.1). The interested reader is referred to our code to see the details.

```
with torch.no_grad():
      delta_norm = delta.norm(p = 2, dim = (2,3))
      delta_index = delta_norm > eps
      delta[delta_index] /=(delta_norm[delta_index][:, None, None]/eps)
```

Simple implementation based on the original DRO approach results in little extra computation complexity, which is also shown in the following track of time during experiments on MNIST dataset.

| Training time per epoch (s) | DRO | Martingale DRO |
|---|---|---|
| Average | 1.66 | 1.73 |
| Variance | $1.90 \times 10^{-3}$ | $2.10 \times 10^{-3}$ |

Table 2: Per-iteration wall-clock time comparison between the vanilla DRO model and Martingale DRO model.

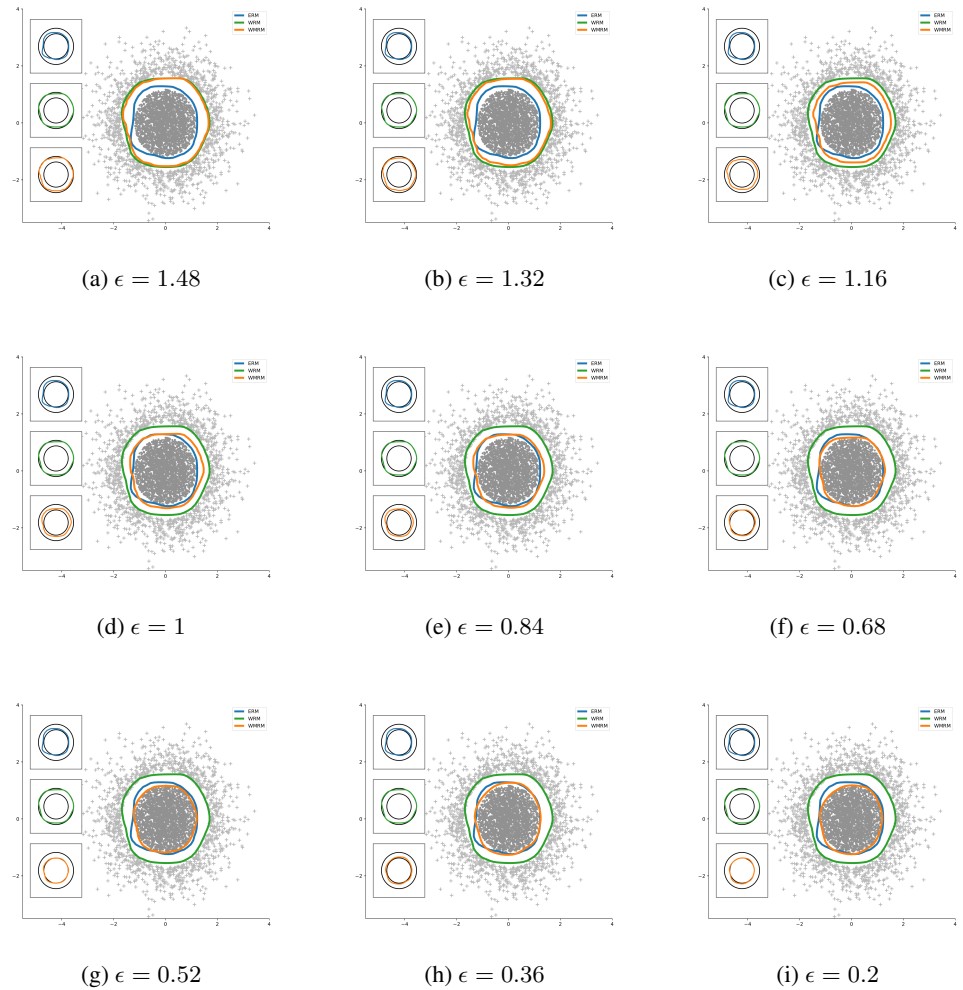

Figure 5: Synthetic data. Decreasing non-zero $\epsilon$'s push the perturbed martingale constraints towards the exact martingale constraints and force the classification boundary increasingly inward.

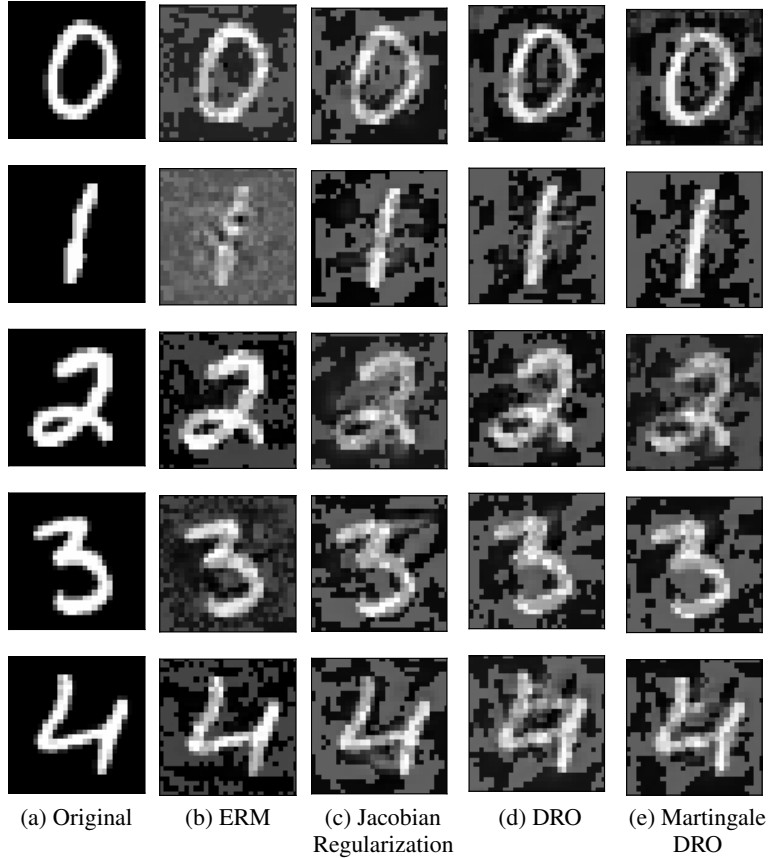

(a) Original     (b) ERM     (c) Jacobian     (d) DRO     (e) Martingale
Regularization           DRO

Figure 6: The largest DRO perturbations such that each model makes correct prediction.

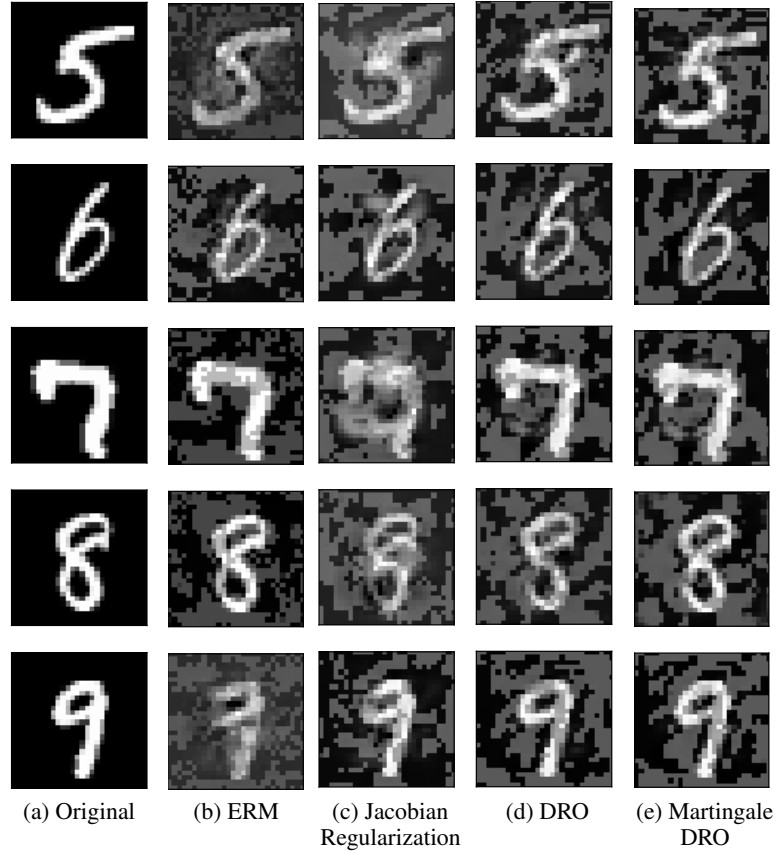

(a) Original     (b) ERM     (c) Jacobian Regularization     (d) DRO     (e) Martingale DRO

Figure 7: The largest DRO perturbations such that each model makes correct prediction.