# OpenReview forum: "Tikhonov Regularization is Optimal Transport Robust under Martingale Constraints"
_NeurIPS.cc/2022/Conference — NeurIPS 2022 Accept_

### Official Review · Reviewer_qbWC · 2022-07-04

**Rating:** 7
**Confidence:** 2
**Soundness:** 4 excellent
**Presentation:** 3 good
**Contribution:** 3 good

**Summary:**

The authors focus on the problem of distributional robust optimization, which consists in training a model, assuming the learned distribution lies in a ball around the empirical dataset (w.r.t. to some metric). They consider the Wasserstein distance as such metric, and propose to add another martingale constraint on the true distribution. It is motivated by imposing a provably higher dispersion on the learned distribution. Inspired by a previous work proving the equivalence between distributionaly robust problem in some setting and a regularized MSE minimization problem, they prove an analog result on their formulation.  The martingale constraint is penalized using a Mahalanobis norm. They propose an subgradient algorithm to estimate the model. They provide experiments on synthetic and MNIST data.

**Questions:**

Could you please provide an answer to the above remarks (1), (3), (4), (5) and (6) ?

**Limitations:**

The authors did not adress societal impact, but I think it is limited for this work.

**Strengths And Weaknesses:**

The contributions of this work are interesting. The equivalence with a regularized MSE problem is interesting. However, I am not an expert of distributionally robust optimization, so I cannot precisely assess the novelty of the authors contributions. I provide below some remarks.

1) The derivation of the duality is a bit fast in the paper. In particular I am not sure how the martingale constraint is handled. It seems that in the proof you assume the adversary distribution is discrete (similar to the empirical distribution), so that the martingale constraint can be handled, and to obtain a duality result as for Theorem 2.3. Is it indeed the case ? If so, isn't it a restriction to assume it is discrete ? The adversary distribution could have a continuous density for instance.

2) There seems to be a typo at line 147.

3) Concerning Proposition 3.6, I wonder how behaves this result when taking asymptotics. For instance if \epsilon goes to infinity, we should retrieve Proposition 2.2, but it does not seem obvious to retrieve the square root of the MSE. Could you discuss why we should retrieve such result ?

4) Could you provide the definition of adversarial RMSE in the main body, so that we are sure what is plotted in figure 1 ?

5) At line 242 you say that "without loss of generality" we can assume that M=I. However I see no argument to justify it. Is it a simplification assumption, or can one prove the loss and gradients is the same for any covariance  M ?

6) It would be interesting to track time in the numerical experiments. The penalty seems much more complicated, thus do the benefits of the performance outweigh the extra computation complexity of the model ?

---

> ### Author Response · Authors · 2022-08-02
> **Response to Reviewer qbWC**
>
> We appreciate and thank the reviewer for their positive comments. We now provide responses to all questions you have raised.
>
> **Q1:** The derivation of the duality is a bit fast in the paper. In particular I am not sure how the martingale constraint is handled. It seems that in the proof you assume the adversary distribution is discrete (similar to the empirical distribution), so that the martingale constraint can be handled, and to obtain a duality result as for Theorem 2.3. Is it indeed the case? If so, isn't it a restriction to assume it is discrete? The adversary distribution could have a continuous density for instance.
>
> **Response:**   We would like to clarify that the only discrete assumption we make is about the nominal distribution $\hat{\mathbb{P}}$: we assume that $\hat{\mathbb{P}}$ is the empirical measure supported on the training data and $\hat{\mathbb{P}}$ is used as the center of the ambiguity set. The adversary can choose any distribution $\mathbb{Q}$ in the ambiguity set containing all Borel probability measures supported on $\mathcal{X}$ that are of a Wasserstein distance less than or equal to $\rho$ from the center distribution $\hat{\mathbb{P}}$. This ambiguity set is non-parametric: it contains continuous, discrete and mixture distributions. We refer the reviewer to line 456--459 in Appendix B for further detailed clarification. Due to the page limit, we cannot include all proof details of the strong duality results in the main context; however, all the proof details can be found in Appendix B.
>
> **Q2:** There seems to be a typo at line 147.
>
> **Response:** There is no typo here. We have discussed the inequality at line 147-148. The inequality holds because the Wasserstein ambiguity set with martingale constraints is a subset of the vanilla Wasserstein ambiguity set (with**out** the martingale constraints). The inequality now follows by basic rules of optimization.
>
> **Q3:** Concerning Proposition 3.6, I wonder how behaves this result when taking asymptotics. For instance if epsilon goes to infinity, we should retrieve Proposition 2.2, but it does not seem obvious to retrieve the square root of the MSE. Could you discuss why we should retrieve such result?
>
> **Response:**
> We refer the reviewer to Theorem 3.7 (line 196). When $\epsilon$ goes to infinity, the minimizer of $s$ in $R(\beta)$ is forced to be $\mathbf{0}$, then equation (3.5) is reduced to ($\gamma = 1$)
> $$
>     \begin{align}
>      \mathcal{L}_\beta(\hat{\mathbb{P}}, \rho, \epsilon) & = \frac{1}{2N} \sum_i (\beta^T X_i)^2 + \frac{\rho \\|\beta\\|_I^2}{2}  + \\|\beta\\|_I \sqrt{\frac{\rho}{N} \sum_i (\beta^TX_i)^2} \\\\
> & = \frac{1}{2}(\sqrt{\frac{1}{N} \sum_i (\beta^TX_i)^2} +\sqrt{\rho}\\|\beta\\|_I)^2.
> \end{align}
> $$
> Here, the identity matrix $I$ can be replaced by the inverse of $M$.
>
> **Q4:** Could you provide the definition of adversarial RMSE in the main body, so that we are sure what is plotted in figure 1?
>
> **Response:**    The adversarial RMSE is defined as
>     $$
>     \textrm{RMSE} = \sqrt{\frac{1}{N}\sum_i (\hat{\beta}^Tx_{adv}^{(i)}-y_{adv}^{(i)})^2},
>     $$
>     where $x_{adv}^{(i)}$ are the generated adversarial samples based on test samples $x_{test}^{(i)}$ via Fast Gradient Method (FGM) and  Projected Gradient Descent (PGD) and $\hat{\beta}$ is the esimator returned by the proposed method.
>
> **Q5:** At line 242 you say that "without loss of generality" we can assume that $M=I$. However I see no argument to justify it. Is it a simplification assumption, or can one prove the loss and gradients is the same for any covariance M?
>
> **Response:** When $M$ is not $I$, then $\\|\beta\\|_{M^{-1}} = \\|M^{-\frac{1}{2}} \beta\\|_2$, which means we just need to insert $M^{-\frac{1}{2}}$ in front of the corresponding $\beta$. We let $M=I$ to make the formula at line 243 to look more clear.
>
> **Q6:**  It would be interesting to track time in the numerical experiments. The penalty seems much more complicated, thus do the benefits of the performance outweigh the extra computation complexity of the model?
>
> **Response:** Thanks for your suggestion. We provide the per-iteration wall-clock time comparison with the vanilla DRO model as below, which shows that the algorithm we propose is fairly efficient and does not cause additional computational burden.
>
> | Training time per epoch (s) | DRO      | Martingale DRO |
> | --------------------------- | -------- | -------------- |
> | Average                     | 1.66     | 1.73           |
> | Variance                    | 1.90E-03 | 2.10E-03       |

---

> > ### Comment · Reviewer_qbWC · 2022-08-08
> > **Author answer to rebuttal**
> >
> > Dear authors,
> > I thank you for your answer which clarified all my points of uncertainty. I think I will keep my score as is.

---

### Official Review · Reviewer_Wc7X · 2022-07-09

**Rating:** 4
**Confidence:** 4
**Soundness:** 3 good
**Presentation:** 2 fair
**Contribution:** 2 fair

**Summary:**

This paper proposes  a connection between Tikhonov regularization and optimal transport map with exact martingale constraints. This finding provides an explanation and guidance to the existing regularization in DRO problem. The authors also introduced a meaningful implementation to solve the problem at hand.

**Questions:**

a)	Theorem 3.7 requires a convex quadratic function and linear mapping. I think these two constraints are quite strict in the adversarial learning problem, how can you guarantee these?  Can one relax these two conditions if so how?
b)	In the first experiment of section 5.2, the  intuitive explanation of the results is not very enlightening if not meaningless to reader. Is there a  statistical measure to estimate the ensuing  results?



**Ethics Review Area:**

["I don’t know"]

**Limitations:**

The model should also be tested on a more complex dataset thereby provide  a deeper understanding on the effectiveness and improvement of the model.

**Strengths And Weaknesses:**

Strengths: The paper  appears to be theoretically sound, firstly establishing a connection between regularization and martingale constraints, thereby building a potential basis for adversarial learning.
Weaknesses: The notation in the paper is rather  hard to track, e.g., L_\beta, l(f_\beta), and f_\beta are three different things. The experiments are rather thin. It  would be more useful  to show some comparisons with existing regularization approaches. Please also see questions below

---

> ### Author Response · Authors · 2022-08-02
> **Response to Reviewer Wc7X**
>
> Thank you for your comments, hopefully the following discussion can clear up your concerns. For the notation issue, we have already tried our best to make all arguments clear and consistent. As the proposed model is indeed the interpolation between empirical risk minimization (with regularization) and the vanilla DRO model, it is necessary to invoke different symbols to distinguish them. We don't believe we introduce superfluous definitions. If you have any other concrete suggestions can help us to further improve it, we are happy to incorporate them in our manuscript. We now provide responses to all questions you have raised.
>
> **Q1:** Theorem 3.7 requires a convex quadratic function and linear mapping. I think these two constraints are quite strict in the adversarial learning problem, how can you guarantee these? Can one relax these two conditions if so how?
>
> **Response:**  Thanks for your question. For the adversarial learning problem, we indeed rely on Proposition 3.6 instead of Theorem 3.7, which is general enough to be applied to deep neural networks. The reformulation is tight without any relaxation here.  In Sections 4.2 and 5.2, we also give the computational scheme and supportive experimental results to corroborate our theoretical findings. To better highlight our theoretical contributions, it would be better to change **Proposition 3.6** to **Theorem 3.6** to let the reader notice that this reformulation result is general enough.
>
>  With this discussion, we would like to re-emphasize our contributions here. First, our paper reveals the equivalence between the Tikhonov regularization and the optimal transport robustification with exact martingale constraints (thus showing that Tikhonov regularization is distributionally robust optimal in a precise sense). Second, the martingale DRO model motivated us to develop a new *perturbed* martingale DRO model which leads to a new class of robustification (or equivalently, a new class of regularization) for the adversarial learning problem, see lines 73-82. We hope that the clarification that we emphasize may provide the reviewer with a better understanding of our contributions.
>
> **Q2:** In the first experiment of section 5.2, the intuitive explanation of the results is not very enlightening if not meaningless to reader. Is there a statistical measure to estimate the ensuing results?
>
> **Response:** We respectfully disagree with the reviewer about the purpose of this experiment. To clarify: our perturbed martingale DRO model is the interpolation between ERM with Tikhonov regularization (with $\epsilon = 0$) and the vanilla optimal transport DRO model (with $\epsilon = \infty$). Moreover, it is well-known that ERM suffers from overfitting while the vanilla optimal transport DRO model may become too over-conservative. We believe that a toy 2-dimensional problem conducted in Figure 2 has explicitly illustrated that our model can alleviate the disadvantages of both extremes and achieve a better model performance.  In Figure 2, it is clear that the classification boundary generated by the vanilla optimal transport DRO (green) misclassifies some outer points (gray) due to its over-conservativeness. Also, the classification boundary generated by ERM with Tikhonov regularization (blue) misclassifies some inner points (darkgray) due to overfitting. We can observe that our proposed perturbed martingale DRO strikes the balance between these two models and achieve 100\% test accuracy. We think this example is quite intuitive to help readers better understand the ``interpolation" effects of the parameter $\epsilon$ (e.g., ERM with Tikhonov regularization at $\epsilon =0$ and vanilla optimal transport DRO at $\epsilon = + \infty$).
>
>  Regarding the statistical measure, as we discussed, our model obtained the higher test accuracy. Besides that, it is clear that the proposed martingale model will have a diagonalize confusion matrix. To further demonstrate the effectiveness of our model, we provide the confusion matrix of three models as below:
>
> | Confusion Matrix |   | ERM  |      | Martingale DRO |      | DRO  |      |
> | ---------------- | -- | ---- | ---- | -------------- | ---- | ---- | ---- |
> | TP               | FN | 1093 | 0    | 1093           | 0    | 1837 | 66   |
> | FP               | TN | 5    | 2902 | 0              | 2907 | 0    | 2907 |

---

> > ### Comment · Reviewer_Wc7X · 2022-08-06
> > **OT_ with_different_flavors**
> >
> > I guess we agree to disagree.
> > I am familiar with both OT and Tikhonov regularization... regularization in OT is meant to reduce the search space by establishing some dependence between the source space and target space.... I don't see that reflected in the provided example example....

---

> > > ### Author Response · Authors · 2022-08-07
> > > **Response to Reviewer Wc7X**
> > >
> > > Thanks for your comments. We may require further clarification from the reviewer in order to answer this question appropriately. What does the reviewer mean about source and target? We just have source data here. We would like to highlight our focus --- OT-based distributionally robust optimization instead of OT itself. It is well known that often OT acts as a function of the gradient norm (i.e., gradient with respect to the data), which is not Tikhonov. Our paper shows that in fact both Tikhonov and Jocabian regularization are optimal in a non-parametric local min-max sense (i.e., distributionally robust). However, we guess you may want to ask about the structure of the adversary and know the difference between the vanilla DRO and the proposed perturbed martingale DRO models. Based on equation (4.1), it is easy to observe that the new adversarial learning paradigm is to further constrain the magnitude of each perturbation no more than eps. We also provide the visualization result to help the reader to get a better understanding of this structure, see Figure 2 (additional one more illustration of the structure of the adversary). As expected when adding our eps budget on perturbation, the Martingale perturbation constrains the magnitude of perturbation to be smaller than eps for each data point, while the original DRO method tends to perturb more wildly. The difference in the structure of the worst-case adversary is more prominent right around the boundaries of the ring regions.
> > >
> > > We sincerely hope you can acknowledge the theoretical merits of this paper and re-evaluate our contributions.  We are also happy to further clarify your concerns during the discussion period.

---

> ### Author Response · Authors · 2022-08-02
> **Response to Reviewer Wc7X Part II**
>
>   **Q3:** The experiments are rather thin. It would be more useful to show some comparisons with existing regularization approaches. Complex dataset?
>
> **Response:** First of all, in Section 5.2, we compared our method with the most relevant regularization technique --- *the Jacobian regularization* on the MNIST dataset,  please refer to Figure 3 and Figure 4 for a detailed comparison of the performance. Furthermore, we would like to emphasize that our paper is a methodology/theory oriented paper: we aim to provide a unified viewpoint of various useful regularizations from the distributionally robust optimization perspective. And use these perspective to obtain a larger class of distributionally robust regularization methods. The experiment results conducted in this paper are to corroborate our theoretical results and lie within the norm of typical theory-focused paper within this conference.  We believe that it is unconventional to expect a theoretically-focused paper to test the model performance on extensive state-of-the-art deep learning models and datasets. Nevertheless, to address your concern, we are happy to further demonstrate the effectiveness of the proposed method on a large dataset --- **CIFAR 10**.
>
> |  PGD Attack | ERM    |  DRO   | Martingale DRO |  Jacobian Regularization |
> | ----------- | ------ | ------ | -------------- | ------------------------ |
> | ϵ = 0       | 84.16% | 84.02% | 85.48%         | 81.73%                   |
> | ϵ = 0.04    | 77.50% | 82.87% | 83.25%         | 78.78%                   |
> | ϵ = 0.08    | 70.20% | 80.68% | 80.86%         | 73.85%                   |
>
> *Experimental Set Up for CIFAR 10* ---  For the classifier, we train a ResNet with the architecture in [1]. We optimize using Adam with a batch size of 128 for all methods. The learning rate starts from 0.01 and shrinks by $0.1^{\frac{\textrm{epoch}}{\textrm{total epochs}}}$,
> and each model is trained for 100 epoches. The simulations are
> implemented using Python 3.8 on Google Colab with TPU v2 and 16 GB RAM.
> Similarily, we test the performance of four methods (ERM, DRO, Jacobian regularization and martingale DRO) under the PGD attack with different levels of perturbation, whose performance is consistent with the results of MNIST. The Top-1 accuracy results are shown in the table above.
>
> [1] He K, Zhang X, Ren S, et al. Deep residual learning for image recognition[C]//Proceedings of the IEEE conference on computer vision and pattern recognition. 2016: 770-778.
>
> Please let us know if our response addresses your concerns. We are happy to address any remaining points during the discussion phase. If our response has adequately addressed your concerns, we kindly ask you to consider raising the score.

---

> > ### Comment · Reviewer_Wc7X · 2022-08-06
> > **Expectations**
> >
> > Why would it be inconventional to expect a theoretical paper to have some substantiation of the claims?
> > That is very common in numerous theoretical papers, and given that significant context may not be present, an illustration can inject additional insight.

---

> > > ### Author Response · Authors · 2022-08-07
> > > **Response to Reviewer  Wc7X**
> > >
> > > Thank you very much for your feedback. Respectfully, we would like to highlight that our paper indeed provides “substantiations” of the claims both from theoretical and empirical perspectives. The effectiveness of the new adversarial training scheme has been validated on real-world datasets **MNIST** and **CIFAR 10** (add it in our first-round rebuttal). Notably, the MNIST dataset is still the field’s standard benchmark dataset to evaluate and compare performance among models to condense and deliver insights. To study how robust a deep learning model is subject to (possibly adversarial) distributional shift, the MNIST dataset is also one of the leading benchmarks [5]. We believe the experiment set up in our paper is aligned with conventions in ML top conferences,e.g., NeurIPS, ICML, and ICLR. Here, we just list several recent theoretical/method-oriented papers published in the last five years to support our claim, see [1-11] for details. Most of them conduct their experiments just on MNIST and few of them provide additional results on a larger dataset --- CIFAR 10. Upon the Reviewer’s suggestions, we have added the experimental results on CIFAR 10 in our first-round response. We observe that the performance is consistent with the result of MNIST. Besides that, we also provide a toy 2D example, see Figure 2 and Figure 5, to illustrate the geometric interpretation of the interpolation between ERM and the vanilla DRO and the structure of adversarial examples.
> > >
> > > Reference:
> > >
> > > [1] Robey, Alexander, et al. "Adversarial robustness with semi-infinite constrained learning." Advances in Neural Information Processing Systems 34 (2021): 6198-6215.
> > >
> > > [2] Jafarpour, Saber, et al. "Robust implicit networks via non-Euclidean contractions." Advances in Neural Information Processing Systems 34 (2021): 9857-9868.
> > >
> > > [3] Lee, Sungyoon, et al. "Towards Better Understanding of Training Certifiably Robust Models against Adversarial Examples." Advances in Neural Information Processing Systems 34 (2021): 953-964.
> > >
> > > [4] Nguyen V A, Zhang F, Blanchet J, et al. Distributionally robust local non-parametric conditional estimation[J]. Advances in Neural Information Processing Systems, 2020, 33: 15232-15242.
> > >
> > > [5] Ovadia Y, Fertig E, Ren J, et al. Can you trust your model's uncertainty? evaluating predictive uncertainty under dataset shift[J]. Advances in neural information processing systems, 2019, 32.
> > >
> > > [6] Wang Y, Jha S, Chaudhuri K. Analyzing the robustness of nearest neighbors to adversarial examples[C]//International Conference on Machine Learning. PMLR, 2018: 5133-5142.
> > >
> > > [7] Bhattacharjee, Robi, and Kamalika Chaudhuri. "When are non-parametric methods robust?." International Conference on Machine Learning. PMLR, 2020.
> > >
> > > [8] Yang, Yao-Yuan, et al. "Robustness for non-parametric classification: A generic attack and defense." International Conference on Artificial Intelligence and Statistics. PMLR, 2020.
> > >
> > > [9] Wong, Eric, and Zico Kolter. "Provable defenses against adversarial examples via the convex outer adversarial polytope." International Conference on Machine Learning. PMLR, 2018.
> > >
> > > [10] Awasthi, Pranjal, Abhratanu Dutta, and Aravindan Vijayaraghavan. "On robustness to adversarial examples and polynomial optimization." Advances in Neural Information Processing Systems 32 (2019).
> > >
> > > [11] Raghunathan, Aditi, Jacob Steinhardt, and Percy S. Liang. "Semidefinite relaxations for certifying robustness to adversarial examples." Advances in Neural Information Processing Systems 31 (2018).

---

> ### Author Response · Authors · 2022-08-05
> **Could you please check our response?**
>
> Dear ReviewerWc7X,
>
> Since only a few days remain in the discussion period, we would appreciate it if you check and reply to our response to your comments soon. This will give us time to address further questions and comments that you may have before the end of the discussion period. If our response adequately addresses your concerns, please consider raising the score of our submission. Thank you very much for your time.
>
> Best,
> The Authors

---

### Official Review · Reviewer_rU1F · 2022-07-11

**Rating:** 7
**Confidence:** 3
**Soundness:** 3 good
**Presentation:** 3 good
**Contribution:** 3 good

**Summary:**

The paper show the equivalence of Optimal Transport Distributional Robust Optimization and Tikhonov regularization under martingale constraints. Inspired by this observation, the paper further introduce some new regularization techniques, both for linear regressions and black-box models such as neural networks.

**Questions:**

1. How do you get the last equality of the proof of Proposition 3.2 in the main text? Can you explain this step clearly? The Holder's inequality provides that
$$ (\beta^{T} \Delta)^2 \leq |\Delta|_{M}^{2}  |\beta|_{M^{-1}}^{2}, $$
with the equality occurs if $\Delta = M^{-1} \beta$ (up to a constants). After that, we need to construct a $\pi$ (or a sequence of $\pi$) to attain this equality and satisfy all the imposed constraints. Although the other longer proof in the Appendix is correct, I am still confused about this short proof in the main text.

2. The fact that we can choose the cost function in OT to be adjusted by a positive definite matrix $M$ is nice. But in practice, how do we choose $M$ rather than identity matrix $I$? It would be helpful to have some sentences to explain it around line 90-93.

**Ethics Review Area:**

["I don’t know"]

**Limitations:**

The paper is well written and has no major limitations.

**Strengths And Weaknesses:**

**Strength**:

1. The presentation in this paper is good, all the proofs in Appendix are carefully written.

2. The proposed regularization method works better than the old OT-DRO method, and the theoretical support is solid.

**Weakness**: I think this is a strong paper with no major weakness.

---

> ### Author Response · Authors · 2022-08-02
> **Response to  Reviewer rU1F**
>
> Thanks for your positive comments. Please find the detailed clarification of your questions as below.
>
> **Q1:** How do you get the last equality of the proof of Proposition 3.2 in the main text? Can you explain this step clearly?
>
> **Response:** For the occurrence of equality, we need to construct $\Delta = c(\Delta) M^{-1} \beta$, $\pi$-a.s., where $c(\Delta) \in \mathbb{R}$ satisfies $E_{\pi}[c(\Delta) | X] = 0$, then a proper scaling will lead to $E_{\pi} [\|\\|\Delta\\|_{M}^2] = \rho$.
>
> **Q2:** The fact that we can choose the cost function in OT to be adjusted by a positive definite matrix  is nice. But in practice, how do we choose  rather than identity matrix? It would be helpful to have some sentences to explain it around line 90-93.
>
> **Response:** Thanks for your suggestion. Usually, the positive definite matrix $M$ is supposed to invoke some prior information. For example, in [3, Section 4.2, equation (24)], the authors tune $M$ using implied volatility to include additional market information in the ambiguity set. Besides, methods developed in metric learning can be also applied to  the selection of $M$. Based on your suggestion, we will definitely include a detailed explanation in the modified version.

---

> > ### Comment · Reviewer_rU1F · 2022-08-05
> > **Response to Q1**
> >
> > Hi, thank you for the answer to my question.
> >
> > How to construct $\Delta$ is exactly what I want to ask. Because $\Delta = c(\Delta) M^{-1} \beta$ here needs to satisfy a lot of constraints. It is the difference of $\overline{X}$ and $X$, where the joint distribution of them is $\pi$. Then it needs to satisfy $E_{\pi}[c(\Delta)|X] = 0$ and $E_{\pi}[c^2(\Delta)] \|\beta\|_{M^{-1}}^2 \leq \rho$. I think it is non-trivial to construct such $c(\Delta)$ and we need more discussion on the proof here.
> >
> > If such a task is non-trivial and you need more space. I think you may change the "Proof of Proposition 3.2." in the main text to a sketch proof. Then refer detailed steps to the Appendix. It would help to clarify the presentation here.

---

> > > ### Author Response · Authors · 2022-08-07
> > > **Response to Reviewer rU1F**
> > >
> > > Thanks for your suggestions. We provide one way to construct $\Delta$. That is , we take a normal random variable $C \sim N(0, \rho)$, which is independent of $X$, and let $\Delta = C M^{-1} \beta$. We have incorporated this detailed explanation into our latest version.

---

### Comment · Area_Chair_NHCb · 2022-08-03
**Discussion period**

Thanks to all reviewers and authors for their work on this submission.

As the discussion period starts, I want to make sure that reviewers have read the author's response, and if needed react to it.

This can be done either by communicating with authors or in private conversation within the reviewing team.

Reviewer Wc7X: Has the author's response appropriately adressed your concerns?

---

> ### Comment · Reviewer_Wc7X · 2022-08-07
> **MY_Rating**
>
> I would change my rating to borderline accept.
> I am just not convinced that the proposed approach is that insightful about OT, as they are missing the viability or new insight to get there, which anyone would expect from a new approach.

---

> > ### Comment · Area_Chair_NHCb · 2022-08-08
> > **Rating update?**
> >
> > Dear reviewer,
> >
> > Thank you very much for getting involved in the discussion and interacting with the reviewers.
> >
> > Are you still considering to update your score?
> >
> > Best,
> > AC

---

> > ### Author Response · Authors · 2022-08-09
> > **Thanks from Authors!**
> >
> > Dear Reviewer Wc7X,
> >
> > Thanks for keeping an open mind and for agreeing to change the score.
> >
> > Following your suggestions regarding the experiments, we have done a new set of experiments that reveals an intriguing qualitative difference in the structure of the adversarial optimal coupling. Please see Figure 2(b) for further details.
> >
> > On the theoretical side, we believe that the equivalence between the Tikhonov regularization and the exact martingale constraint in OT-based DRO is well-motivated. Upon this interesting hidden connection, it is natural for us to develop the perturbed martingale DRO model to get rid of the disadvantages of standard regularization techniques and the over-conservative vanilla DRO model.
> >
> > Overall, our paper has significant theoretical and modeling contributions. As the discussion will be closed soon, we would like to take this last opportunity to address your follow-up concerns or questions.
> >
> > Thank you for your consideration!
> >
> >
> > All the best,
> > Authors

---

### Meta-Review · Area_Chair_NHCb · 2022-08-26

**Recommendation:** Accept
**Confidence:** Certain

**Metareview:**

This work focuses on robust stochastic optimization (under a Wasserstein constraint), and shows the efficiency of Tikhonov regularization for this problem.

There has been a lively and constructive discussion between authors and reviewers, and ultimately all agree that this work should be accepted, and so do I.

**Award:**

No

---

### Decision · Program_Chairs · 2022-09-14

Accept